# ANCHOR & TRANSFORM:
# LEARNING SPARSE EMBEDDINGS FOR LARGE VOCABULARIES

**Paul Pu Liang**$^{\heartsuit\spadesuit *}$, **Manzil Zaheer**$^{\heartsuit}$, **Yuan Wang**$^{\heartsuit}$, **Amr Ahmed**$^{\heartsuit}$
$^{\heartsuit}$Google Research, $^{\spadesuit}$Carnegie Mellon University
pliang@cs.cmu.edu, {manzilzaheer,yuanwang,amra}@google.com

## ABSTRACT

Learning continuous representations of discrete objects such as text, users, movies, and URLs lies at the heart of many applications including language and user modeling. When using discrete objects as input to neural networks, we often ignore the underlying structures (e.g., natural groupings and similarities) and embed the objects independently into individual vectors. As a result, existing methods do not scale to large vocabulary sizes. In this paper, we design a simple and efficient embedding algorithm that learns a small set of anchor embeddings and a sparse transformation matrix. We call our method ANCHOR & TRANSFORM (ANT) as the embeddings of discrete objects are a sparse linear combination of the anchors, weighted according to the transformation matrix. ANT is scalable, flexible, and end-to-end trainable. We further provide a statistical interpretation of our algorithm as a Bayesian nonparametric prior for embeddings that encourages sparsity and leverages natural groupings among objects. By deriving an approximate inference algorithm based on Small Variance Asymptotics, we obtain a natural extension that automatically learns the optimal number of anchors instead of having to tune it as a hyperparameter. On text classification, language modeling, and movie recommendation benchmarks, we show that ANT is particularly suitable for large vocabulary sizes and demonstrates stronger performance with fewer parameters (up to $40\times$ compression) as compared to existing compression baselines. Code for our experiments can be found at `https://github.com/pliang279/sparse_discrete`.

## 1 INTRODUCTION

Most machine learning models, including neural networks, operate on vector spaces. Therefore, when working with discrete objects such as text, we must define a method of converting objects into vectors. The standard way to map objects to continuous representations involves: 1) defining the *vocabulary* $V = \{v_1, ..., v_{|V|}\}$ as the set of all objects, and 2) learning a $|V| \times d$ *embedding matrix* that defines a $d$ dimensional continuous representation for each object. This method has two main shortcomings. Firstly, when $|V|$ is large (e.g., million of words/users/URLs), this embedding matrix does not scale elegantly and may constitute up to $80\%$ of all trainable parameters (Jozefowicz et al., 2016). Secondly, despite being discrete, these objects usually have underlying structures such as natural groupings and similarities among them. Assigning each object to an individual vector assumes independence and foregoes opportunities for statistical strength sharing. As a result, there has been a large amount of interest in learning *sparse interdependent representations* for large vocabularies rather than the full embedding matrix for cheaper training, storage, and inference.

In this paper, we propose a simple method to learn sparse representations that uses a global set of vectors, which we call the *anchors*, and expresses the embeddings of discrete objects as a *sparse linear combination* of these anchors, as shown in Figure 1. One can consider these anchors to represent latent topics or concepts. Therefore, we call the resulting method ANCHOR & TRANSFORM (ANT). The approach is reminiscent of low-rank and sparse coding approaches, however, surprisingly in the literature these methods were not elegantly integrated with deep networks. Competitive attempts are often complex (e.g., optimized with RL (Joglekar et al., 2019)), involve multiple training stages (Ginart et al., 2019; Liu et al., 2017), or require post-processing (Svenstrup et al., 2017; Guo et al., 2017; Aharon et al., 2006; Awasthi & Vijayaraghavan, 2018). We derive a simple optimization objective which learns these anchors and sparse transformations in an end-to-end manner. ANT is

---

$^{*}$work done during an internship at Google.

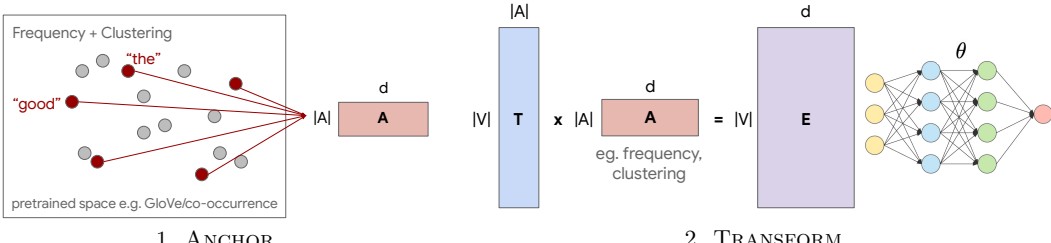

Figure 1: ANCHOR & TRANSFORM (ANT) consists of two components: 1) ANCHOR: Learn embeddings $\mathbf{A}$ of a small set of anchor vectors $A = \{a_1, ..., a_{|A|}\}, |A| << |V|$ that are *representative* of all discrete objects, and 2) TRANSFORM: Learn a sparse transformation $\mathbf{T}$ from the anchors to the full embedding matrix $\mathbf{E}$. $\mathbf{A}$ and $\mathbf{T}$ are trained end-to-end for specific tasks. ANT is scalable, flexible, and allows the user to easily incorporate domain knowledge about object relationships. We further derive a Bayesian nonparametric view of ANT that yields an extension NBANT which automatically tunes $|A|$ to achieve a balance between performance and compression.

scalable, flexible, and allows the user flexibility in defining these anchors and adding more constraints on the transformations, possibly in a domain/task specific manner. We find that our proposed method demonstrates stronger performance with fewer parameters (up to $40\times$ compression) on multiple tasks (text classification, language modeling, and recommendation) as compared to existing baselines.

We further provide a statistical interpretation of our algorithm as a Bayesian nonparametric (BNP) prior for neural embeddings that encourages sparsity and leverages natural groupings among objects. Specifically, we show its equivalence to Indian Buffet Process (IBP; Griffiths & Ghahramani (2005)) prior for embedding matrices. While such BNP priors have proven to be a flexible tools in graphical models to encourage hierarchies (Teh & Jordan, 2010), sparsity (Knowles & Ghahramani, 2011), and other structural constraints (Roy et al., 2016), these inference methods are usually complex, hand designed for each setup, and non-differentiable. Our proposed method opens the door towards integrating priors (e.g., IBP) with neural representation learning. These theoretical connections leads to practical insights - by asymptotically analyzing the likelihood of our model in the small variance limit using Small Variance Asymptotics (SVA; Roweis (1998)), we obtain a natural extension, NBANT, that automatically learns the optimal number of anchors to achieve a balance between performance and compression instead of having to tune it as a hyperparameter.

## 2 RELATED WORK

Prior work in learning sparse embeddings of discrete structures falls into three categories:

**Matrix compression techniques** such as low rank approximations (Acharya et al., 2019; Grachev et al., 2019; Markovsky, 2011), quantizing (Han et al., 2016), pruning (Anwar et al., 2017; Dong et al., 2017; Wen et al., 2016), or hashing (Chen et al., 2015; Guo et al., 2017; Qi et al., 2017) have been applied to embedding matrices. However, it is not trivial to learn *sparse* low-rank representations of large matrices, especially in conjunction with neural networks. To the best of our knowledge, we are the first to present the integration of sparse low-rank representations, their non-parametric extension, and demonstrate its effectiveness on many tasks in balancing the tradeoffs between performance & sparsity. We also outperform many baselines based on low-rank compression (Grachev et al., 2019), sparse coding (Chen et al., 2016b), and pruning (Liu et al., 2017).

**Reducing representation size:** These methods reduce the dimension $d$ for different objects. Chen et al. (2016a) divides the embedding into buckets which are assigned to objects in order of importance, Joglekar et al. (2019) learns $d$ by solving a discrete optimization problem with RL, and Baevski & Auli (2019) reduces dimensions for rarer words. These methods resort to RL or are difficult to tune with many hyperparameters. Each object is also modeled independently without information sharing.

**Task specific methods** include learning embeddings of only common words for language modeling (Chen et al., 2016b; Luong et al., 2015), and vocabulary selection for text classification (Chen et al., 2019). Other methods reconstruct pre-trained embeddings using codebook learning (Chen et al., 2018; Shu & Nakayama, 2018) or low rank tensors (Sedov & Yang, 2018). However, these methods cannot work for general tasks. For example, methods that only model a subset of objects cannot be used for retrieval because it would never retrieve the dropped objects. Rare objects might be highly relevant to a few users so it might not be ideal to completely ignore them. Similarly, task-specific methods such as subword (Bojanowski et al., 2017) and wordpiece (Wu et al., 2016) embeddings, while useful for text, do not generalize to general applications such as item and query retrieval.

## 3 ANCHOR & TRANSFORM

Suppose we are presented with data $\mathbf{X} \in V^N, \mathbf{Y} \in \mathbb{R}^{N \times c}$ drawn from some joint distribution $p(x, y)$, where the support of $x$ is over a discrete set $V$ (the vocabulary) and $N$ is the size of the training set. The entries in $\mathbf{Y}$ can be either discrete (classification) or continuous (regression). The goal is to learn a $d$-dimensional *representation* $\{\mathbf{e}_1, ..., \mathbf{e}_{|V|}\}$ for each object by learning an embedding matrix $\mathbf{E} \in \mathbb{R}^{|V| \times d}$ where row $i$ is the representation $\mathbf{e}_i$ of object $i$. A model $f_\theta$ with parameters $\theta$ is then used to predict $y$, i.e., $\hat{y}_i = f_\theta(x_i; \mathbf{E}) = f_\theta(\mathbf{E}[x_i])$.

At a high level, to encourage statistical sharing between objects, we assume that the embedding of each object is obtained by linearly superimposing a small set of anchor objects. For example, when the objects considered are words, the anchors may represent latent abstract concepts (of unknown cardinality) and each word is a weighted mixture of different concepts. More generally, the model assumes that there are some unknown number of anchors, $A = \{\mathbf{a}_1, ..., \mathbf{a}_{|A|}\}$. The embedding $\mathbf{e}_i$ for object $i$ is generated by first choosing whether the object possesses each anchor $\mathbf{a}_k \in \mathbb{R}^d$. The selected anchors then each contribute some weight to the representation of object $i$. Therefore, instead of learning the large embedding matrix $\mathbf{E}$ directly, ANT consists of two components:

1) ANCHOR: Learn embeddings $\mathbf{A} \in \mathbb{R}^{|A| \times d}$ of a small set of anchor objects $A = \{\mathbf{a}_1, ..., \mathbf{a}_{|A|}\}, |A| << |V|$ that are *representative* of all discrete objects.

2) TRANSFORM: Learn a sparse transformation $\mathbf{T}$ from $\mathbf{A}$ to $\mathbf{E}$. Each of the discrete objects is *induced* by some transformation from (a few) anchor objects. To ensure sparsity, we want $\text{nnz}(\mathbf{T}) << |V| \times d$.

$\mathbf{A}$ and $\mathbf{T}$ are trained end-to-end for task specific representations. To enforce sparsity, we use an $\ell_1$ penalty on $\mathbf{T}$ and constrain its domain to be non-negative to reduce redundancy in transformations (positive and negative entries canceling out).

**Algorithm 1** ANCHOR & TRANSFORM algorithm for learning sparse representations of discrete objects.

**ANCHOR & TRANSFORM:**
1: Anchor: initialize anchor embeddings $\mathbf{A}$.
2: Transform: initialize $\mathbf{T}$ as a *sparse matrix*.
3: Optionally + domain info: initialize domain sparsity matrix $\mathbf{S}(G)$ as a *sparse matrix* (see Appendix F).
4: **for** each batch $(\mathbf{X}, \mathbf{Y})$ **do**
5:     Compute loss $\mathcal{L} = \sum_i D_\phi(y_i, f_\theta(x_i; \mathbf{TA}))$
6:     $\mathbf{A}, \mathbf{T}, \theta = \text{UPDATE}(\nabla\mathcal{L}, \eta)$.
7:     $\mathbf{T} = \max\{(\mathbf{T} - \eta\lambda_2) \odot \mathbf{S}(G) + \mathbf{T} \odot (1 - \mathbf{S}(G)), 0\}$.
8: **end for**
9: **return** anchor embeddings $\mathbf{A}$ and transformations $\mathbf{T}$.

$$\min_{\mathbf{T} \geq 0, \, \mathbf{A}, \theta} \sum_i D_\phi(y_i, f_\theta(x_i; \mathbf{TA})) + \lambda_2 \|\mathbf{T}\|_1, \tag{1}$$

where $D_\phi$ is a suitable Bregman divergence between predicted and true labels, and $\|\mathbf{T}\|_1$ denotes the sum of absolute values. Most deep learning frameworks directly use subgradient descent to solve eq (1), but unfortunately, such an approach will not yield sparsity. Instead, we perform optimization by proximal gradient descent (rather than approximate subgradient methods which have poorer convergence around non-smooth regions, e.g., sparse regions) to ensure exact zero entries in $\mathbf{T}$:

$$\mathbf{A}^{t+1}, \mathbf{T}^{t+1}, \theta^{t+1} = \text{UPDATE}\left(\nabla \sum_i D_\phi(y_i, f_\theta(x_i; \mathbf{T}^t\mathbf{A}^t)), \eta\right), \tag{2}$$

$$\mathbf{T}^{t+1} = \text{PROX}_{\eta\lambda_2}(\mathbf{T}^{t+1}) = \max\left(\mathbf{T}^{t+1} - \eta\lambda_2, 0\right), \tag{3}$$

where $\eta$ is the learning rate, and UPDATE is a gradient update rule (e.g., SGD (Lecun et al., 1998), ADAM (Kingma & Ba, 2015), YOGI (Zaheer et al., 2018)). $\text{PROX}_{\eta\lambda_2}$ is a composition of two proximal operators: 1) soft-thresholding (Beck & Teboulle, 2009) at $\eta\lambda_2$ which results from subgradient descent on $\lambda_2\|\mathbf{T}\|_1$, and 2) $\max(\cdot, 0)$ due to the non-negative domain for $\mathbf{T}$. We implement this proximal operator on top of the YOGI optimizer for our experiments.

Together, equations (2) and (3) give us an iterative process for end-to-end learning of $\mathbf{A}$ and $\mathbf{T}$ along with $\theta$ for specific tasks (Algorithm 1). $\mathbf{T}$ is implemented as a *sparse matrix* by only storing its non-zero entries and indices. Since $\text{nnz}(\mathbf{T}) << |V| \times d$, this makes storage of $\mathbf{T}$ extremely efficient as compared to traditional approaches of computing the entire $|V| \times d$ embedding matrix. We also provide implementation tips to further speedup training and ways to incorporate ANT with existing speedup techniques like softmax sampling (Mikolov et al., 2013) or noise-contrastive estimation (Mnih & Teh, 2012) in Appendix H. After training, we only store $|A| \times d + \text{nnz}(\mathbf{T}) << |V| \times d$ entries that define the complete embedding matrix, thereby using fewer parameters than the traditional $|V| \times d$ matrix. General purpose matrix compression techniques such as hashing (Qi et al., 2017), pruning (Dong

et al., 2017), and quantizing (Han et al., 2016) are compatible with our method: the matrices $\mathbf{A}$ and $\text{nnz}(\mathbf{T})$ can be further compressed and stored.

We first discuss practical methods for anchor selection (§3.1). In Appendix F we describe several ways to incorporate domain knowledge into the anchor selection and transform process. We also provide a statistical interpretation of ANT as a sparsity promoting generative process using an IBP prior and derive approximate inference based on SVA (§3.2). This gives rise to a nonparametric version of ANT that automatically learns the optimal number of anchors.

## 3.1 ANCHOR: SELECTING THE ANCHORS $A$

Inspired by research integrating initialization strategies based on clustering (Teh et al., 2007) and Coresets (Bachem et al., 2015) with Bayesian nonparametrics, we describe several practical methods to select anchor objects that are most representative of all objects (refer to Appendix D for a comparison of initialization strategies.).

**Frequency and TF-IDF:** For tasks where frequency or TF-IDF (Ramos, 1999) are useful for prediction, the objects can simply be sorted by frequency and the most common objects selected as the anchor points. While this might make sense for tasks such as language modeling (Luong et al., 2015; Chen et al., 2016b), choosing the most frequent objects might not cover rare objects that are not well represented by common anchors.

**Clustering:** To ensure that all objects are close to some anchor, we use $k$-means++ initialization (Arthur & Vassilvitskii, 2007). Given a feature space representative of the relationships between objects, such as Glove (Pennington et al., 2014) for words or a co-occurrence matrix (Haralick et al., 1973) for more general objects, $k$-means++ initialization picks cluster centers to span the entire space. This can augment other strategies, such as initializing anchors using frequency followed by clustering to complete remaining anchors (see Figure 2).

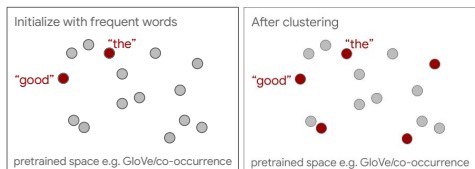

Figure 2: An illustration of initialization strategies for anchors combining ideas from frequency and $k$-means++ clustering initialization. Clustering initialization picks anchors to span the space of all objects after frequent objects have been selected.

**Random basis vectors:** Initialize $\mathbf{A}$ to a set of random basis vectors. This simple yet powerful method captures the case where we have less knowledge about the objects (i.e., without access to any pretrained representation/similarity space).

## 3.2 STATISTICAL INTERPRETATION AS A BAYESIAN NONPARAMETRIC PRIOR

To provide a statistical interpretation of ANT, we first analyze a generative process for discrete representations that is consistent with our algorithm. Given a set of anchors, $A = \{\mathbf{a}_1, ..., \mathbf{a}_{|A|}\}$, we use a binary latent variable $z_{ik} \in \{0, 1\}$ to indicate whether object $i$ possesses anchor $k$ and a positive latent variable $w_{ik} \in \mathbb{R}_{\geq 0}$ to denote the weight that anchor $k$ contributes towards object $i$. Therefore, the representation $\mathbf{e}_i$ is given by $\mathbf{e}_i = \sum_k w_{ik} z_{ik} \mathbf{a}_k$. Ideally, we want the vector $\mathbf{z}_i$ to be sparse for efficient learning and storage. More formally, suppose there are $K := |A|$ anchors, then:

- $\mathbf{Z} \in \mathbb{R}^{|V| \times K} \sim \text{IBP}(a, b);\ \mathbf{A} \in \mathbb{R}^{K \times d} \sim P(\mathbf{A}) = \mathcal{N}(0, 1);\ \mathbf{W} \in \mathbb{R}^{|V| \times K} \sim P(\mathbf{W}) = \text{Exp}(1)$
- for $i = 1, \cdots, N$
  - $\hat{y}_i = f_\theta(x_i; (\mathbf{Z} \circ \mathbf{W})\mathbf{A})$
  - $y_i \sim p(y_i | x_i; \mathbf{Z}, \mathbf{W}, \mathbf{A}) = \exp\{-D_\phi(y_i, \hat{y}_i)\} b_\phi(y_i)$

In this generative process, the selection matrix $\mathbf{Z}$ follows a two-parameter Indian Buffet Process (IBP; Griffiths & Ghahramani (2005)) prior (Ghahramani et al., 2007). Not only does this BNP prior allow for a potentially infinite number of anchors, but it also encourages each object to only select a small subset of anchors, resulting in a sparse $\mathbf{z}_i$ (see Appendix A for details). We place a standard Gaussian prior on the continuous anchors embeddings $\mathbf{a}_k$ and an exponential prior on the weights $\mathbf{W}$ which give the actual non-negative transformation weights for the non-zero entries defined in $\mathbf{Z}$. $\mathbf{E} = (\mathbf{Z} \circ \mathbf{W})\mathbf{A}$ is the final embedding learnt by our model which represents a $d$-dimensional continuous *representation* $\{\mathbf{e}_1, ..., \mathbf{e}_{|V|}\}$ for each discrete object where row $i$ is the representation $\mathbf{e}_i$ of object $i$. Finally, a neural model $f_\theta$ with parameters $\theta$ is used to predict $y_i$ given the embedded representations, i.e., $\hat{y}_i = f_\theta(x_i; (\mathbf{Z} \circ \mathbf{W})\mathbf{A}) = f_\theta((\mathbf{Z} \circ \mathbf{W})\mathbf{A}[x_i])$.

**Likelihood Model/Loss:** We assume that the final emission model $y_i|\hat{y}_i$ belongs to the exponential family. Since exponential family distributions have a corresponding Bregman divergence (Banerjee et al. (2005); see Appendix C for examples), we choose $D_\phi(y_i, \hat{y}_i)$ as the corresponding Bregman divergence between predicted and true labels. Appropriate choices for $D_\phi$ recover cross-entropy and MSE losses. $b_\phi(y_i)$ does not depend on any learnable parameter or variable and can be ignored.

**Joint likelihood:** Under the generative model as defined above, the joint likelihood is given by:

$$\log p(\mathbf{Y}, \mathbf{Z}, \mathbf{W}, \mathbf{A}|\mathbf{X}) \propto \sum_i \log p(y_i|x_i; \mathbf{Z}, \mathbf{W}, \mathbf{A}) + \log p(\mathbf{Z}) + \log p(\mathbf{W}) + \log p(\mathbf{A})$$

$$= \sum_i \left\{ -D_\phi(y_i, f_\theta(x_i; (\mathbf{Z} \circ \mathbf{W})\mathbf{A})) + \log b_\phi(y_i) \right\} + \log p(\mathbf{Z}) + \log p(\mathbf{W}) + \log p(\mathbf{A}).$$

However, calculating the posterior or MAP estimate is hard, especially due to the presence of the non-linear deep network $f_\theta$. Approximate inference methods such as MCMC, variational inference, or probabilistic programming would be computationally and statistically inefficient since it would involve sampling, evaluating, or training the model multiple times. To tackle this problem, we perform approximate inference via Small Variance Asymptotics (SVA), which captures the benefits of rich latent-variable models while providing a framework for scalable optimization (Broderick et al., 2013a; Jiang et al., 2012; Roychowdhury et al., 2013).

**Approximate Inference via SVA:** To use SVA, we introduce a scaling variable $\beta$ and shrink the variance of the emission probability by taking $\beta \to \infty$. The scaled probability emission becomes

$$p(y_i|x_i; \mathbf{Z}, \mathbf{W}, \mathbf{A}) = \exp\left\{-\beta D_\phi(y_i, \hat{y}_i)\right\} b_{\beta\phi}(y_i). \tag{4}$$

Following Broderick et al. (2013a), we modulate the number of features in the large-$\beta$ limit by choosing constants $\lambda_1 > \lambda_2 > 0$ and setting the IBP hyperparameters $a = \exp(-\beta\lambda_1)$ and $b = \exp(\beta\lambda_2)$. This prevents a limiting objective function that favors a trivial cluster assignment (every data point assigned to its own separate feature). Maximizing the asymptotic joint likelihood (after taking limits, i.e., $\lim_{\beta\to\infty} \frac{1}{\beta} \log p(\mathbf{Y}, \mathbf{Z}, \mathbf{W}, \mathbf{A}|\mathbf{X})$) results in the following objective function:

$$\min_{\mathbf{T} \geq 0, \ \mathbf{A}, \theta, K} \sum_i D_\phi(y_i, f_\theta(x_i; \mathbf{TA})) + \lambda_2 \|\mathbf{T}\|_0 + (\lambda_1 - \lambda_2)K, \tag{5}$$

where we have combined the variables $\mathbf{Z}$ and $\mathbf{W}$ with their constraints into one variable $\mathbf{T}$. The exponential prior for $\mathbf{W}$ results in a non-negative domain for $\mathbf{T}$. Please refer to Appendix B for derivations. Note that eq (5) suggests a natural objective function in learning representations that minimize the prediction loss $D_\phi(y_i, f_\theta(x_i; \mathbf{TA}))$ while ensuring sparsity of $\mathbf{T}$ as measured by the $\ell_0$-norm and using as few anchors as possible ($K$). Therefore, optimizing eq (5) gives rise to a nonparametric version of ANT, which we call NBANT, that automatically learns the optimal number of anchors. To perform optimization over the number of anchors, our algorithm starts with a small $|A| = 10$ and either adds anchors (i.e., adding a new row to $\mathbf{A}$ and a new column to $\mathbf{T}$) or deletes anchors to minimize eq (5) at every epoch depending on the trend of the objective evaluated on validation set. We outline the exact algorithm in Appendix G along with more implementation details.

Analogously, we can derive the finite case objective function for a fixed number of anchors $K$:

$$\min_{\mathbf{T} \geq 0, \ \mathbf{A}, \theta} \sum_i D_\phi(y_i, f_\theta(x_i; \mathbf{TA})) + \lambda_2 \|\mathbf{T}\|_0, \tag{6}$$

which, together with a $\ell_1$ penalty on $\mathbf{T}$ as a convex relaxation for the $\ell_0$ penalty, recovers the objective function in eq (1). The solution for this finite version along with $K$ yields the Pareto front. Different values of $\lambda_1$ in eq (5) can be used for model selection along the front as elucidated in Appendix L.

# 4 EXPERIMENTS

To evaluate ANT, we experiment on text classification, language modeling, and movie recommendation tasks. Experimental details are in Appendix J and full results are in Appendix K.

## 4.1 TEXT CLASSIFICATION

**Setup:** We follow the setting in Chen et al. (2019) with four datasets: AG-News ($V = 62K$) (Zhang et al., 2015), DBPedia ($V = 563K$) (Lehmann et al., 2015), Sogou-News ($V = 254K$) (Zhang et al., 2015), and Yelp-review ($V = 253K$) (Zhang et al., 2015). We use a CNN for classification (Kim, 2014). ANT is used to replace the input embedding and domain knowledge is derived from WordNet and co-occurrence in the training set. We record test accuracy and number of parameters used in the embedding only. For ANT, num params is computed as $|A| \times d + \text{nnz}(\mathbf{T})$.

Table 1: Text classification results on AG-News. Our approach with different initializations achieves within 0.5% accuracy with 40× fewer parameters, outperforming the published compression baselines. Init: initialization method, Acc: accuracy, # Emb: number of (non-zero) embedding parameters.

| Method | $|A|$ | Init $A$ | Sparse $\mathbf{T}$ | $\mathbf{T} \geq 0$ | Acc (%) | # Emb (M) |
|---|---|---|---|---|---|---|
| CNN (Zhang et al., 2015) | 61, 673 | All | ✗ | ✗ | 91.6 | 15.87 |
| FREQUENCY (Chen et al., 2019) | 5, 000 | Frequency | ✗ | ✗ | 91.0 | 1.28 |
| TF-IDF (Chen et al., 2019) | 5, 000 | TF-IDF | ✗ | ✗ | 91.0 | 1.28 |
| GL (Chen et al., 2019) | 4, 000 | Group lasso | ✗ | ✗ | 91.0 | 1.02 |
| VVD (Chen et al., 2019) | 3, 000 | Var dropout | ✗ | ✗ | 91.0 | 0.77 |
| SPARSEVD (Chirkova et al., 2018) | 5, 700 | Mult weights | ✗ | ✗ | 88.8 | 1.72 |
| SPARSEVD-VOC (Chirkova et al., 2018) | 2, 400 | Mult weights | ✗ | ✗ | 89.2 | 0.73 |
| SPARSE CODE (Chen et al., 2016b) | 100 | Frequency | ✓ | ✗ | 89.5 | 2.03 |
| ANT | 50 | Frequency | ✓ | ✓ | 89.5 | 1.01 |
| | 10 | Frequency | ✓ | ✓ | **91.0** | **0.40** |
| | 10 | Random | ✓ | ✓ | 90.5 | **0.40** |

**Baselines:** On top of the **CNN**, we compare to the following compression approaches. Vocabulary selection methods: 1) **FREQUENCY** where only embeddings for most frequent words are learnt (Chen et al., 2016b; Luong et al., 2015), 2) **TF-IDF** which only learns embeddings for words with high TF-IDF score (Ramos, 1999), 3) **GL** (group lasso) which aims to find underlying sparse structures in the embedding matrix via row-wise $\ell_2$ regularization (Liu et al., 2015; Park et al., 2016; Wen et al., 2016), 4) **VVD** (variational vocabulary dropout) which performs variational dropout for vocabulary selection (Chen et al., 2019). We also compare to 5) **SPARSEVD** (sparse variational dropout) which performs variational dropout on all parameters (Chirkova et al., 2018), 6) **SPARSEVD-VOC** which uses multiplicative weights for vocabulary sparsification (Chirkova et al., 2018), and 7) a **SPARSE CODE** model that learns a sparse code to reconstruct pretrained word representations (Chen et al., 2016b). All CNN architectures are the same for all baselines with details in Appendix J.1.

**Results** on AG-News are in Table 1 and results for other datasets are in Appendix K.1. We observe that restricting $\mathbf{T} \geq 0$ using an exponential prior is important in reducing redundancy in the entries. Domain knowledge from WordNet and co-occurrence also succeeded in reducing the total (non-zero) embedding parameters to 0.40M, a compression of 40× and outperforming the existing approaches.

## 4.2 LANGUAGE MODELING

**Setup:** We perform experiments on word-level Penn Treebank (PTB) ($V = 10K$) (Marcus et al., 1993) and WikiText-103 ($V = 267K$) (Merity et al., 2017) with LSTM (Hochreiter & Schmidhuber, 1997) and AWD-LSTM (Merity et al., 2018). We use ANT as the input embedding tied to the output embedding. Domain knowledge is derived from WordNet and co-occurrence on the training set. We record the test perplexity and the number of (non-zero) embedding parameters.

**Baselines:** We compare to **SPARSEVD** and **SPARSEVD-VOC**, as well as low-rank (**LR**) and tensor-train (**TT**) model compression techniques (Grachev et al., 2019). Note that the application of variational vocabulary selection to language modeling with tied weights is non-trivial since one is unable to predict next words when words are dynamically dropped out. We also compare against methods that compress the trained embedding matrix as a *post-processing* step before evaluation: **POST-SPARSE HASH** (post-processing using sparse hashing) (Guo et al., 2017) and **POST-SPARSE HASH+$k$-SVD** (Awasthi & Vijayaraghavan, 2018; Guo et al., 2017) which uses $k$-SVD (which is the basis of dictionary learning/sparse coding) (Aharon et al., 2006) to solve for a sparse embedding matrix, instead of adhoc-projection in (Guo et al., 2017). Comparing to these post-processing methods demonstrates that end-to-end training of sparse embeddings is superior to post-compression.

**Results:** On PTB (Table 2), we improve the perplexity and compression as compared to previously proposed methods. We observe that sparsity is important: baseline methods that only perform lower-rank compression with dense factors (e.g., LR LSTM) tend to suffer in performance and use many parameters, while ANT retains performance with much better compression. ANT also outperforms post-processing methods (POST-SPARSE HASH), we hypothesize this is because these post-processing methods accumulate errors in both language modeling as well as embedding reconstruction. Using an anchor size of $500/1,000$ reaches a good perplexity/compression trade-off: we reach within 2 points perplexity with 5× reduction in parameters and within 7 points perplexity with 10× reduction. Using AWD-LSTM, ANT with $1,000$ dynamic basis vectors is able to compress parameters by 10× while achieving 72.0 perplexity. Incorporating domain knowledge allows us to further compress the parameters by *another* 10× and achieve 70.0 perplexity, which results in 100× total compression.

Table 2: Language modeling on PTB (**top**) and WikiText-103 (**bottom**). We outperform existing vocabulary selection, low-rank, tensor-train, and post-compression (hashing) baselines on performance and compression metrics. Ppl: perplexity, # Emb: number of (non-zero) embedding parameters.

| Method (PTB) | $\|A\|$ | Init $A$ | Sparse $\mathbf{T}$ | $\mathbf{T} \geq 0$ | Ppl | # Emb (M) |
|---|---|---|---|---|---|---|
| LSTM (Chirkova et al., 2018) | $10,000$ | All | ✗ | ✗ | 70.3 | 2.56 |
| LR LSTM (Grachev et al., 2019) | $10,000$ | All | ✗ | ✗ | 112.1 | 1.26 |
| TT LSTM (Grachev et al., 2019) | $10,000$ | All | ✗ | ✗ | 116.6 | 1.16 |
| AWD-LSTM (Merity et al., 2018) | $10,000$ | All | ✗ | ✗ | 59.0 | 4.00 |
| SPARSEVD (Chirkova et al., 2018) | $9,985$ | Mult weights | ✗ | ✗ | 109.2 | 1.34 |
| SPARSEVD-VOC (Chirkova et al., 2018) | $4,353$ | Mult weights | ✗ | ✗ | 120.2 | 0.52 |
| POST-SPARSE HASH (Guo et al., 2017) | $1,000$ | Post-processing | ✓ | ✗ | 118.8 | 0.60 |
| POST-SPARSE HASH+$k$-SVD | $1,000$ | Post-processing | ✓ | ✗ | 78.0 | 0.60 |
| ANT | $2,000$ | Random | ✓ | ✓ | **71.5** | 0.78 |
| | $1,000$ | Random | ✓ | ✓ | 73.1 | 0.49 |
| | $100$ | Random | ✓ | ✓ | 96.5 | **0.05** |
| | $100$ | Frequency | ✓ | ✓ | **70.0** | **0.05** |

| Method (WikiText-103) | $\|A\|$ | Init $A$ | Sparse $\mathbf{T}$ | $\mathbf{T} \geq 0$ | Ppl | # Emb (M) |
|---|---|---|---|---|---|---|
| AWD-LSTM (Merity et al., 2018) | $267,735$ | All | ✗ | ✗ | 35.2 | 106.8 |
| HASH EMBED (Svenstrup et al., 2017) | $10,000$ | Frequency | ✗ | ✗ | 70.2 | 4.4 |
| POST-SPARSE HASH (Guo et al., 2017) | $1,000$ | Post-processing | ✓ | ✗ | 764.7 | 5.7 |
| POST-SPARSE HASH+$k$-SVD | $1,000$ | Post-processing | ✓ | ✗ | 73.7 | 5.7 |
| ANT | $1,000$ | Random ($\lambda_2 = 1 \times 10^{-6}$) | ✓ | ✓ | **38.4** | 6.5 |
| | $500$ | Random ($\lambda_2 = 1 \times 10^{-5}$) | ✓ | ✓ | 54.2 | **0.4** |

On WikiText-103, we train using sampled softmax (Bengio & Senecal, 2008) (due to large vocabulary) for $500,000$ steps. To best of our knowledge, we could not find literature on compressing language models on WikiText-103. We tried general compression techniques like low rank tensor and tensor train factorization (Grachev et al., 2019), but these did not scale. As an alternative, we consider a **HASH EMBED** baseline that retains the frequent $k$ words and hashes the remaining words into $1,000$ OOV buckets (Svenstrup et al., 2017). We vary $k \in \{1 \times 10^5, 5 \times 10^4, 1 \times 10^4\}$ (details in Appendix J.3). From Table 2 (bottom), we reach within 3 perplexity with ~ $16\times$ reduction in parameters and within 13 perplexity with ~ $80\times$ reduction, outperforming the frequency and hashing baselines. We observe that ANT's improvement over post-compression methods (POST-SPARSE HASH) is larger on WikiText than PTB, suggesting that ANT is particularly suitable for large vocabularies.

## 4.3 RECOMMENDER SYSTEMS

**Setup:** We perform experiments on both movie and product recommendation tasks. For movie recommendations, we follow Ginart et al. (2019) and we experiment on MovieLens 25M (Harper & Konstan, 2015) with 126K users and 59K movies. We also present results for MovieLens 1M in Appendix K.3. On product recommendation, we show that ANT scales to Amazon Product reviews (Ni et al., 2019), the largest existing dataset for recommender systems with 233M reviews spanning 43.5M users and 15.2M products. Following Wan et al. (2020), we ensured that the users and products in the test set have appeared in the training data for generalization.

**Baselines:** We compare to a baseline Matrix Factorization (**MF**) model (Koren et al., 2009) with full embedding matrices for movies and users and to Mixed Dimension (**MIXDIM**) embeddings (Ginart et al., 2019), a compression technique that assigns different dimension to different users/items based on popularity. We also compare to **SPARSE CBOW** (Sun et al., 2016) which learns sparse $\mathbf{E}$ by placing an $\ell_1$ penalty over all entries of $\mathbf{E}$ and optimizing using online subgradient descent, and **SLIMMING** (Liu et al., 2017), which performs subgradient descent before pruning small weights by setting them to 0. Such methods learn embeddings for objects independently without statistical strength sharing among related objects. We also test NBANT using the algorithm derived from the Bayesian nonparametric interpretation of ANT.

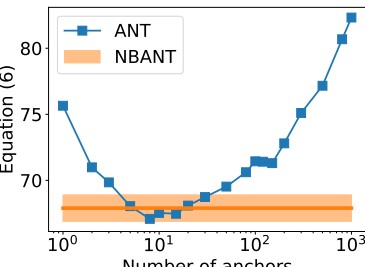

Figure 3: NBANT accurately selects the optimal $\|A\|$ to minimize eq (5), thereby achieving a balance between performance and compression.

**Results:** From Table 3, ANT outperforms standard matrix factorization and dense mixed dimensional embeddings for performance and compression. NBANT is also able to automatically select an optimal

Table 3: On Movielens 25M, ANT outperforms MF and mixed dimensional embeddings. NBANT automatically tunes $|A|$ ($^*$denotes $|A|$ discovered by NBANT) to achieve a balance between performance and compression.

| Method | user $|A|$ | item $|A|$ | Init $A$ | MSE | # Emb (M) |
|---|---|---|---|---|---|
| MF (Koren et al., 2009) | 162K | 59K | All | 0.665 | 3.55 |
| MixDim (Ginart et al., 2019) | 162K | 59K | All | 0.662 | 0.89 |
| Sparse CBOW (Sun et al., 2016) | 162K | 59K | Random ($\lambda = 1 \times 10^{-6}$) | 0.695 | 2.44 |
| Sparse CBOW (Sun et al., 2016) | 162K | 59K | Random ($\lambda = 2 \times 10^{-6}$) | 0.757 | 1.45 |
| Slimming (Liu et al., 2017) | 162K | 59K | Random ($\lambda = 2 \times 10^{-6}$) | 0.735 | 1.89 |
| Slimming (Liu et al., 2017) | 5 | 5 | Random ($\lambda = 2 \times 10^{-6}$) | 0.656 | 0.97 |
| ANT | 10 | 15 | Random ($\lambda_2 = 2 \times 10^{-6}$) | **0.617** | 1.76 |
| | 5 | 5 | Random ($\lambda_2 = 2 \times 10^{-6}$) | 0.651 | **0.89** |
| NBANT | Auto $\to 8^*$ | Auto $\to 8^*$ | Random ($\lambda_1 = 0.1, \lambda_2 = 2 \times 10^{-6}$) | 0.637 | 1.29 |
| | Auto $\to 6^*$ | Auto $\to 6^*$ | Random ($\lambda_1 = 0.01, \lambda_2 = 2 \times 10^{-6}$) | 0.649 | 1.09 |

Table 4: ANT scales to Amazon Product reviews, the largest existing dataset for recommender systems with 233M reviews spanning 43.5M users and 15.2M products, and is able to perform well under compression.

| Data | Method | user $|A|$ | item $|A|$ | Init $A$ | MSE | # Emb (M) |
|---|---|---|---|---|---|---|
| Electronics | MF ($d = 10$, Wan et al. 2020) | 9.84M | 0.76M | All | 1.590 | 105 |
| | MF | 9.84M | 0.76M | All | 1.524 | 170 |
| | ANT | 20 | 8 | Random ($\lambda_2 = 5 \times 10^{-7}$) | **1.422** | 25.8 |
| | | 8 | 3 | Random ($\lambda_2 = 1 \times 10^{-6}$) | 1.529 | 7.10 |
| | | 5 | 3 | Random ($\lambda_2 = 2 \times 10^{-6}$) | 1.591 | **3.89** |
| All | MF | 43.5M | 15.2M | All | 1.164 | 939 |
| | ANT | 15 | 10 | Random ($\lambda_2 = 1 \times 10^{-7}$) | **1.099** | 201 |
| | | 8 | 8 | Random ($\lambda_2 = 1 \times 10^{-7}$) | 1.167 | **95.9** |

number of anchors (6/8) to achieve solutions along the performance-compression Pareto front. In Figure 3, we plot the value of eq (5) across values of $|A|$ after a comprehensive hyperparameter sweep on ANT across 1000 settings. In comparison, NBANT optimizes $|A|$ and reaches a good value of eq (5) *in a single run* without having to tune $|A|$ as a hyperparameter, thereby achieving best balance between performance and compression. Please refer to Appendix K.3 for more results and discussion on NBANT.

For product recommendation, we first experiment on a commonly used subset of the data, Amazon Electronics (with 9.84M users and 0.76M products), to ensure that our results match published baselines (Wan et al., 2020), before scaling our experiment to the entire dataset. From Table 4, we find that ANT compresses embeddings by 25× on Amazon Electronics while maintaining performance, and 10× on the full Amazon reviews dataset.

**Online NBANT:** Since NBANT automatically grows/contracts $|A|$ during training, we can further extend NBANT to an online version that sees a stream of batches without revisiting previous ones (Bryant & Sudderth, 2012). We treat each batch as a new set of data coming in and train on that batch until convergence, modify $|A|$ as in Algorithm 2, before moving onto the next batch. In this significantly more challenging online setting, NBANT is still able to learn well and achieve a MSE of 0.875 with 1.25M non zero parameters. Interestingly this online version of NBANT settled on a similar range of final user (8) and item (8) anchors as compared to the non-online version (see Table 3), which confirms the robustness of NBANT in finding relevant anchors automatically. In Appendix K.3 we discuss more observations around online NBANT including ways of learning $|A|$.

## 4.4 Discussion and Observations

Here we list some general observations regarding the importance of various design decisions in ANT:

1) **Sparsity is important:** Baselines that compress with dense factors (e.g., LR, TT) suffer in performance while using many parameters, while ANT retains performance with better compression.

2) **Choice of $A$:** We provide results on more clustering initializations in Appendix D. In general, performance is robust w.r.t. choice of $A$. While frequency and clustering work better, using a dynamic basis also performs well. Thus, it is beneficial to use any extra information about the discrete objects (e.g., domain knowledge or having a good representation space like GloVe to perform clustering).

Table 5: Word association results after training language models with ANT on the word-level PTB dataset. **Left**: the non-anchor words most induced by a given anchor word. **Right**: the largest (non-anchor, anchor) entries learnt in **T** after sparse $\ell_1$-regularization. **Bottom**: movie clusters obtained by sorting movies with the highest coefficients with each anchor embedding.

| Largest word pairs |
| :---: |
| *trading, brokerage* |
| *stock, junk* |
| *year, summer* |
| *york, angeles* |
| *year, month* |
| *government, administration* |

| Anchor words | Non-anchor words |
| :---: | :---: |
| *year* | *august, night, week, month, monday, summer, spring* |
| *stock* | *bonds, certificates, debt, notes, securities, mortgages* |

| Movies | Genre |
| :---: | :---: |
| *God's Not Dead, Sex and the City, Sex and the City 2, The Twilight Saga: Breaking Dawn - Part 1, The Princess Diaries 2: Royal Engagement, The Last Song, Legally Blonde 2: Red, White & Blonde, The Twilight Saga: Eclipse, Maid in Manhattan, The Twilight Saga: Breaking Dawn - Part 2* | *romance, comedy* |
| *Nostalghia, Last Days, Chimes at Midnight, Lessons of Darkness, Sonatine, Band of Outsiders, Gerry, Cyclo, Mishima: A Life in Four Chapters, George Washington* | *drama, indie* |

3) **Anchors and sparse transformations learned:** We visualize the important transformations (large entries) learned between anchors and non-anchors in Table 5. Left, we show the most associated non-anchors for a given anchor word and find that the induced non-anchors are highly plausible: *stock* accurately contributes to *bonds, certificates, securities*, and so on. Right, we show the largest (non-anchor, anchor) pairs learned, where we find related concepts such as *(billion, trillion)* and *(government, administration)*. On MovieLens, for each anchor, we sort the movies according to the magnitude of their transformation coefficients which *automatically discovers movie clusters based on underlying genres*. We obtain a genre purity ratio of $61.7\%$ by comparing automatically discovered movie clusters with the true genre tags provided in MovieLens.

4) **Zero transformations learned:** For MovieLens, we find that ANT assigns 2673 out of 59047 movies to an entire zero row, of which $84\%$ only had 1 rating (i.e., very rare movies). Therefore, compression automatically discovers very rare objects (1 labeled point). On WikiText-103, rare words (e.g., *Anarky, Perl, Voorhis, Gaudí, Lat, Bottomley, Nescopeck*) are also automatically assigned zero rows when performing high compression (54.2 ppl with 0.4M params). Certain rare words that might be predictive, however, are assigned non-zero rows in **T**, such as: *sociologists, deadlines, indestructible, causeways, outsourced, glacially, heartening, unchallenging, roughest*.

5) **Choice of $\lambda_1, \lambda_2$:** Tuning $\lambda_1$ allows us to perform model selection by controlling the trade-off between $|A|$ (model complexity) and performance. By applying eq (5) on our trained models in Table 2, choosing a small $\lambda_1 = 2 \times 10^{-5}$ prefers more anchors ($|A| = 1,000$) and better performance (ppl = 79.4), while a larger $\lambda_1 = 1 \times 10^{-1}$ selects fewer anchors ($|A| = 100$) with a compromise in performance (ppl = 106.6). Tuning $\lambda_2$ allows us to control the tradeoff between sparsity and performance (see details in Appendix L).

6) **Convergence:** In Figure 4, we plot the empirical convergence of validation loss across epochs. ANT converges as fast as the (non-sparse) MF baseline, and faster than compression baselines MixDim (Ginart et al., 2019) and Sparse CBOW (Sun et al., 2016). ANT also converges to the best validation loss.

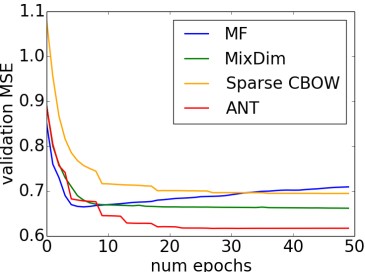

Figure 4: ANT converges faster and to a better validation loss than the baselines.

7) **Scalability:** In addition to fast convergence, ANT also works effectively on large datasets such as Movielens 25M (162K users, 59K movies, 25M examples) and WikiText-103 (267K unique words, 103M tokens). For each epoch on Movielens 25M, standard MF takes 165s on a GTX 980 Ti GPU while ANT takes 176s for $|A| = 5$ and 180s for $|A| = 20$. ANT also scales to the largest recommendation dataset, Amazon reviews, with 25M users and 9M products.

## 5 CONCLUSION

This paper presented ANCHOR & TRANSFORM to learn sparse embeddings of large vocabularies using a small set of anchor embeddings and a sparse transformation from anchors to all objects. We also showed a statistical interpretation via integrating IBP priors with neural representation learning. Asymptotic analysis of the likelihood using SVA yields an extension that automatically learns the optimal number of anchors. On text classification, language modeling, and recommender systems, ANT outperforms existing approaches with respect to accuracy and sparsity.

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

APPENDIX

# A  INDIAN BUFFET PROCESS WITH TWO PARAMETERS

In this section we provide a more detailed treatment of the Indian Buffet Process (IBP) (Griffiths & Ghahramani, 2005; 2011; Paisley et al., 2010) as well its two-parameter generalization (Ghahramani et al., 2007). We begin with describing the single parameter version, which essentially defines a probability distribution over sparse binary matrices with a finite number of rows and an unbounded number of columns. Under IBP prior with hyperparameter $a$, to generate such a sparse binary random matrix $\mathbf{Z}$ with $|V|$ rows, we have the following process:

$$
v_j \sim \mathrm{Beta}(a, 1)
$$

$$
b_k = \prod_{j=1}^{k} v_j \tag{7}
$$

$$
z_{ik} \sim \mathrm{Bernoulli}(b_k), \; i = 1, ..., |V|
$$

It can be shown from this construction that a given matrix $\mathbf{Z}$ will have non-zero probability under the IBP prior if and only if the number of columns containing non-zero entries is finite, albeit a random quantity (Griffiths & Ghahramani, 2005). Also note that $b_k$ keeps diminishing as $0 < v_j < 1$, thus most $z_{ik}$ will be 0, thereby giving rise to the desired sparsity. Moreover, it can be shown that number of number of non-empty columns would be $O(\log |V|)$ in expectation.

Like most Bayesian nonparametric models, it is best understood with an analogy. Consider of a set of customers (objects) queued up to take dishes (features/anchors) from a buffet arranged in a line. The first customer starts at beginning of the buffet and takes a serving of all of the first Poisson$(a)$ dishes. The remaining customers are more picky. The $i$th customer samples dishes in proportion to their popularity (i.e., with probability $m_k/i$), where $m_k$ is the number of previous customers who have sampled a dish. Having reached the end of all previous sampled dishes, the $i$th customer then tries Poisson$(a/i)$ new dishes. The result of this process for the entire vocabulary $V$ is a binary matrix $\mathbf{Z}$ with $|V|$ rows and infinitely many columns where $z_{ik} = 1$ if the $i$th customer sampled the $k$th dish.

Using either description of the IBP, we can find the distribution of the sparse binary matrix $\mathbf{Z}$ with $|V|$ rows and $K$ non-empty columns to be given by (Griffiths & Ghahramani, 2005; Broderick et al., 2013b):

$$
p(\mathbf{Z}) = \frac{a^K}{\prod_{h=1}^{2^{|V|}-1} K_h!} \exp(-a H_{|V|}) \prod_{k=1}^{K} \frac{(|V| - m_k)! (m_k - 1)!}{|V|!}, \tag{8}
$$

where $m_k$ denotes number of customers (objects) who selected dish (anchor) $k$, $H_{|V|}$ is the $|V|$-th Harmonic number $H_{|V|} = \sum_{j=1}^{|V|} \frac{1}{j}$, and $K_h$ is the number of occurrences of the non-zero binary vector $h$ among the columns in $\mathbf{Z}$.

However, the number of features per object and the total number of features are directly coupled through the single parameter $a$. The two-parameter generalization of the IBP allows one to independently tune the average number of features for each object and the overall number of features used across all $V$ objects (Ghahramani et al., 2007). In particular, we now have an additional hyperparameter $b$ along with $a$. The first customer, as before, samples Poisson$(a)$ dishes. However, the $i$-th customer now samples in proportion to their popularity with probability $m_k/(i + b)$, where $m_k$ is the number of previous customers who have sampled a dish. Having reached the end of all previously sampled dishes, the $i$th customer tries Poisson$(ab/(i + b))$ new dishes. The parameter $b$ is introduced in such a way as to preserve the expected number of features per object to be still $a$, but the expected overall number of features is now $O(ab \log |V|)$. The total number of features used thus increases as $b$ increases, thus providing more control on sparsity of $\mathbf{Z}$. This two parameter IBP prior for a binary matrices $\mathbf{Z}$ with $|V|$ rows and $K$ non-empty columns is given by:

$$
p(\mathbf{Z}) = \frac{(ab)^K}{\prod_{h=1}^{2^{|V|}-1} K_h!} \exp(-ab H_{|V|}) \prod_{k=1}^{K} \frac{\Gamma(m_k) \Gamma(|V| - m_k + b)}{\Gamma(|V| + b)} \tag{9}
$$

where $m_k$ denotes number of customers (objects) who selected dish (anchor) $k$ and $H_{|V|} = \sum_{j=1}^{|V|} \frac{1}{b+j-1}$. This distribution is suitable for use as a prior for $\mathbf{Z}$ in models that represent objects using a sparse but potentially infinite array of features.

Historically, IBP has been used as a prior in latent feature models, where the binary entries $z_{ik}$ of a random matrix encode whether feature $k$ is used to explain observation $i$. The IBP can be further

combined with a simple observation model $p(y_i|z_{i,:}, \theta)$ from the exponential family like the Gaussian distribution (see e.g. Griffiths & Ghahramani (2011)). The complexity of inference, using MCMC or variational methods, for such binary factor analysis models has limited the adoption of more complicated observation models. In this work, we break this barrier and, to the best of our knowledge, are the first to integrate IBP with deep representation learning of discrete objects by employing an efficient SVA based inference. Thus, our approach combines the representation capabilities of neural networks with desirable sparsity properties.

## B   DERIVATION OF OBJECTIVE FUNCTION VIA SVA

In this section we derive our objective function using Small Variance Asymptotics (SVA) (Jiang et al., 2012). Recall that the generative process in our model is given by:

- $\mathbf{Z} \in \mathbb{R}^{|V| \times K} \sim \text{IBP}(a, b)$
- $\mathbf{A} \in \mathbb{R}^{K \times d} \sim P(\mathbf{A}) = \mathcal{N}(0, 1)$
- $\mathbf{W} \in \mathbb{R}^{|V| \times K} \sim P(\mathbf{W}) = \text{Exponential}(1)$
- for $i = 1, \cdots, N$
  - $\hat{y}_i = f_\theta(x_i; (\mathbf{Z} \circ \mathbf{W})\mathbf{A})$
  - $y_i \sim p(y_i|x_i; \mathbf{Z}, \mathbf{W}, \mathbf{A}) = \exp\{-D_\phi(y_i, \hat{y}_i)\} b_\phi(y_i)$

The joint log-likelihood under our generative model above is therefore:

$$\log p(\mathbf{Y}, \mathbf{Z}, \mathbf{W}, \mathbf{A}|\mathbf{X})$$
$$\propto \sum_i \log p(y_i|x_i, \mathbf{Z}, \mathbf{W}, \mathbf{A}) + \log p(\mathbf{Z}) + \log p(\mathbf{W}) + \log p(\mathbf{A})$$
$$= \sum_i \{-D_\phi(y_i, f_\theta(x_i, (\mathbf{Z} \circ \mathbf{W})\mathbf{A})) + \log b_\phi(y_i)\} + \log p(\mathbf{Z}) + \log p(\mathbf{W}) + \log p(\mathbf{A}). \quad (10)$$

To use SVA, an approximate objective function for finding point estimates is obtained by taking the limit of the emission probability variances down to zero. We begin by introducing a scaling variable $\beta$ and shrinking the variance of the emission probability to 0 by taking $\beta \to \infty$. The scaled probability emission becomes

$$p(y_i|x_i, \mathbf{Z}, \mathbf{W}, \mathbf{A}) = \exp\{-\beta D_\phi(y_i, \hat{y}_i)\} b_{\beta\phi}(y_i) \quad (11)$$

Following Broderick et al. (2013a), we modulate the number of features in the large-$\beta$ limit by choosing constants $\lambda_1 > \lambda_2 > 0$ and setting the IBP hyperparameters with $\beta$ as follows:

$$a = \exp(-\beta\lambda_1) \qquad b = \exp(\beta\lambda_2) \quad (12)$$

This prevents a limiting objective function that favors a trivial cluster assignment (every data point assigned to its own separate feature).

We now take the limit of the log-likelihood term by term:

$$\lim_{\beta \to \infty} \frac{1}{\beta} \log p(\mathbf{Y}, \mathbf{A}, \mathbf{W}, \mathbf{Z}|\mathbf{X}) \quad (13)$$
$$= \lim_{\beta \to \infty} \frac{1}{\beta} \log p(y_i|x_i, \mathbf{Z}, \mathbf{W}, \mathbf{A}) + \lim_{\beta \to \infty} \frac{1}{\beta} \log p(\mathbf{Z}) + \lim_{\beta \to \infty} \frac{1}{\beta} \log p(\mathbf{W}) + \lim_{\beta \to \infty} \frac{1}{\beta} \log p(\mathbf{A}). \quad (14)$$

- $\lim_{\beta \to \infty} \frac{1}{\beta} \log p(y_i|x_i, \mathbf{Z}, \mathbf{W}, \mathbf{A})$
  $= \lim_{\beta \to \infty} \frac{1}{\beta} (-\beta D_\phi(y_i, \hat{y}_i) + \log b_{\beta\phi}(y_i))$
  $= -D_\phi(y_i, \hat{y}_i) + O(1).$
- $\lim_{\beta \to \infty} \frac{1}{\beta} \log p(\mathbf{Z}) = -\lambda_2 \|\mathbf{Z}\|_0 - (\lambda_1 - \lambda_2)K$, see box below.
- $\lim_{\beta \to \infty} \frac{1}{\beta} \log p(\mathbf{W}) = 0$, if $\mathbf{W} \geq 0$ else $-\infty$.
- $\lim_{\beta \to \infty} \frac{1}{\beta} \log p(\mathbf{A}) = 0$ as $\log p(\mathbf{A}) = O(1)$.

For convenience, we re-write the limit of the IBP prior as

$$\lim_{\beta \to \infty} \frac{1}{\beta} \log p(Z) = \underbrace{\lim_{\beta \to \infty} \frac{1}{\beta} \log \frac{(ab)^K}{\prod_{h=1}^{2^{|V|-1}} K_h}}_{\text{ⓐ}}$$

$$+ \underbrace{\lim_{\beta \to \infty} \frac{1}{\beta} \log \exp(-abH_{|V|})}_{\text{ⓑ}} \tag{15}$$

$$+ \sum_{k=1}^{K} \underbrace{\lim_{\beta \to \infty} \frac{1}{\beta} \log \frac{\Gamma(m_k)\Gamma(|V| - m_k + b)}{\Gamma(|V| + b)}}_{\text{ⓒ}}$$

For part ⓐ:

$$\begin{aligned}
\lim_{\beta \to \infty} \frac{1}{\beta} \log \frac{(ab)^K}{\prod_{h=1}^{2^{|V|-1}} K_h} &= \lim_{\beta \to \infty} \frac{1}{\beta} \log \frac{\exp(-\beta(\lambda_1 - \lambda_2)K)}{\prod_{h=1}^{2^{|V|-1}} K_h} \\
&= \lim_{\beta \to \infty} \frac{1}{\beta} \times -\beta(\lambda_1 - \lambda_2)K - \lim_{\beta \to \infty} \frac{1}{\beta} \times O(1) \\
&= -(\lambda_1 - \lambda_2)K
\end{aligned} \tag{16}$$

For part ⓑ:

$$\begin{aligned}
\lim_{\beta \to \infty} \frac{1}{\beta} \log \exp(-abH_{|V|}) &= \lim_{\beta \to \infty} \frac{1}{\beta} \times -abH_{|V|} \\
&= \lim_{\beta \to \infty} \frac{-\exp(-\beta(\lambda_1 - \lambda_2)K)}{\beta} \times \sum_{j=1}^{|V|} \frac{1}{\exp(\beta\lambda_2) + j - 1} \\
&= 0
\end{aligned} \tag{17}$$

For part ⓒ:

$$\begin{aligned}
\lim_{\beta \to \infty} \frac{1}{\beta} \log \frac{\Gamma(m_k)\Gamma(|V| - m_k + b)}{\Gamma(|V| + b)} &= \lim_{\beta \to \infty} \frac{1}{\beta} \log \Gamma(m_k) - \lim_{\beta \to \infty} \frac{1}{\beta} \sum_{j=1}^{m_k} \log(|V| - j + b) \\
&= 0 - \sum_{j=1}^{m_k} \lim_{\beta \to \infty} \frac{\log(|V| - j + \exp(\beta\lambda_2))}{\beta} \\
&= -\sum_{j=1}^{m_k} \lambda_2 \\
&= -\lambda_2 m_k
\end{aligned} \tag{18}$$

We know that $m_k$ is the number of objects which uses anchor $k$ which counts the number of non-zero entries in the $k$-th column of $\mathbf{Z}$. When we sum over all $k$, it just becomes the number of non-zero entries in $\mathbf{Z}$, which is equivalent to the $L_0$ norm of $\mathbf{Z}$, i.e., $\|\mathbf{Z}\|_0$.

Therefore, the MAP estimate under SVA as given by

$$\max \lim_{\beta \to \infty} \frac{1}{\beta} \log p(\mathbf{Y}, \mathbf{A}, \mathbf{W}, \mathbf{Z}|\mathbf{X}) \tag{19}$$

is equivalent to optimizing the following objective function:

$$\max_{\substack{\mathbf{Z} \in 0,1 \\ \mathbf{W} \geq 0 \\ \mathbf{A}, \theta, K}} \sum_i -D_\phi(y_i, f_\theta(x_i, (\mathbf{Z} \circ \mathbf{W})\mathbf{A})) - \lambda_2 \|\mathbf{Z}\|_0 - (\lambda_1 - \lambda_2)K, \tag{20}$$

where the exponential prior for $\mathbf{W}$ resulted in a limiting domain for $\mathbf{W}$ to be positive. Note that we can combine the optimizing variables $\mathbf{Z}$ and $\mathbf{W}$ with their constraints into one variable $\mathbf{T} \geq 0$. Also

we can switch from a maximization problem to a minimization problem by absorbing the negative sign. Finally we arrive at the desired objective:

$$\min_{\substack{\mathbf{T} \geq 0 \\ \mathbf{A}, \theta, K}} \sum_i D_\phi(y_i, f_\theta(x_i, \mathbf{TA})) + \lambda_2 \|\mathbf{T}\|_0 + (\lambda_1 - \lambda_2)K. \tag{21}$$

## C  EXPONENTIAL FAMILY DISTRIBUTIONS AS BREGMAN DIVERGENCES

In this section we provide some results that relate exponential families distributions and Bregman divergences. As a result, we can relate likelihood models from Sec. 3.2 to appropriate Bregman divergences. Thus, a probabilistic observation model can be translated to a loss functions minimizing the Bregman divergence, which are more amenable to deep network training using gradient based methods. We begin by defining the Bregman divergence below and stating the relationship formally in Theorem 1.

**Definition 1.** *(Bregman, 1967) Let $\phi : S \rightarrow \mathbb{R}$, $S = dom(\phi)$ be a strictly convex function defined on a convex set $S \subset \mathbb{R}^d$ such that $\phi$ is differentiable on $ri(S)$, assumed to be non-empty. The Bregman divergence $D_\phi : S \times ri(S) \rightarrow [0, \infty)$ is defined as*

$$D_\phi(\mathbf{x}, \mathbf{y}) = \phi(\mathbf{x}) - \phi(\mathbf{y}) - \langle \mathbf{x} - \mathbf{y}, \nabla\phi(\mathbf{y}) \rangle, \tag{22}$$

*where $\nabla\phi(y)$ represents the gradient vector of $\phi$ evaluated at $\mathbf{y}$.*

**Theorem 1.** *(Banerjee et al., 2005) There is a bijection between regular exponential families and regular Bregman divergences. In particular, for any exponential family distribution $p(\mathbf{x}|\boldsymbol{\theta}) = p_0(\mathbf{x})\exp(\langle \mathbf{x}, \boldsymbol{\theta} \rangle - g(\boldsymbol{\theta}))$ can be written as $p(\mathbf{x}|\boldsymbol{\mu}) = \exp(-D_\phi(\mathbf{x}, \boldsymbol{\mu}))b_\phi(\mathbf{x})$ where $\phi$ is the Legendre dual of the log-partition function $g(\boldsymbol{\theta})$ and $\boldsymbol{\mu} = \nabla_{\boldsymbol{\theta}} g(\boldsymbol{\theta})$.*

From Theorem 1, we can see that maximizing log-likelihood $\log p(\mathbf{x}|\boldsymbol{\theta})$ is same as minimizing the Bregman divergence $D_\phi(\mathbf{x}, \boldsymbol{\mu})$. Note that we can ignore $b_\phi(\mathbf{x})$ as it depends only on observed data and does not depend on any parameters. We now illustrate some common examples of exponential families (like Gaussian and categorical), derive their corresponding Bregman divergences, and connect to usual loss functions used in deep networks (like MSE and cross-entropy).

**Example 1: Gaussian distribution.** (Banerjee et al., 2005) We start with the unit variance spherical Gaussian distributions with with mean $\boldsymbol{\mu}$, which have densities of the form:

$$p(\mathbf{x}; \boldsymbol{\mu}) = \frac{1}{\sqrt{(2\pi)^d}} \exp\left(-\frac{1}{2}\|\mathbf{x} - \boldsymbol{\mu}\|_2^2\right). \tag{23}$$

Using the log-partition function for Gaussian distribution, we can calculate that $\phi(x) = \frac{1}{2}\|x\|^2$, which yields Bregman divergence equal to:

$$D_\phi(\mathbf{x}, \boldsymbol{\mu}) = \phi(\mathbf{x}) - \phi(\boldsymbol{\mu}) - \langle \mathbf{x} - \boldsymbol{\mu}, \nabla\phi(\boldsymbol{\mu}) \rangle \tag{24}$$

$$= \frac{1}{2}\|\mathbf{x}\|_2^2 - \frac{1}{2}\|\boldsymbol{\mu}\|_2^2 - \langle \mathbf{x} - \boldsymbol{\mu}, \boldsymbol{\mu} \rangle \tag{25}$$

$$= \underbrace{\frac{1}{2}\|\mathbf{x} - \boldsymbol{\mu}\|_2^2}_{\text{mean squared error}}, \tag{26}$$

Thus, $D_\phi(\mathbf{x}, \boldsymbol{\mu})$ along with constant $b_\phi(\mathbf{x})$ given by

$$b_\phi(\mathbf{x}) = \frac{1}{\sqrt{(2\pi)^d}}, \tag{27}$$

recovers the Gaussian density $p(\mathbf{x}) = \exp(-D_\phi(\mathbf{x}, \boldsymbol{\mu}))b_\phi(\mathbf{x})$. Therefore, when we assume that labels have a Gaussian emmission model, the corresponding Bregman divergence $D_\phi(\mathbf{x}, \boldsymbol{\mu}) = \frac{1}{2}\|\mathbf{x} - \boldsymbol{\mu}\|_2^2$ recovers the squared loss commonly used for regression.

**Example 2: Multinomial distribution.** (Banerjee et al., 2005) Another exponential family that is widely used is the family of multinomial distributions:

$$p(\mathbf{x}, \mathbf{q}) = \frac{N!}{\prod_{j=1}^d x_j!} \prod_{j=1}^d q_j^{x_j} \tag{28}$$

where $x_j \in \mathbb{Z}^+$ are frequencies of events, $\sum_{j=1}^d x_j = N$ and $q_j \geq 0$ are probabilities of events, $\sum_{j=1}^d q_j = 1$. The multinomial density can be expressed as the density of an exponential distribution

in $\mathbf{x} = \{x_j\}_{j=1}^{d-1}$ with natural parameter $\boldsymbol{\theta} = \log\left(\frac{q_j}{q_d}\right)_{j=1}^{d-1}$, cumulant function $g(\boldsymbol{\theta}) = -N\log q_d$, and expectation parameter $\boldsymbol{\mu} = \nabla g(\boldsymbol{\theta}) = [Nq_j]_{j=1}^{d-1}$. The Legendre dual $\phi$ of $g$ is given by

$$\phi(\boldsymbol{\mu}) = N\sum_{j=1}^{d}\left(\frac{\mu_j}{N}\right)\log\left(\frac{\mu_j}{N}\right) = N\sum_{j=1}^{d} q_j \log q_j. \tag{29}$$

As a result, the multinomial density can be expressed as a Bregman divergence equal to:

$$D_\phi(\mathbf{x}, \boldsymbol{\mu}) = \underbrace{\sum_{j=1}^{d} x_j \log x_j}_{\text{constant}} - \underbrace{\sum_{j=1}^{d} x_j \log \mu_j}_{\text{cross-entropy loss}}. \tag{30}$$

and constant $b_\phi(\mathbf{x})$ given by

$$b_\phi(\mathbf{x}) = \frac{\prod_{j=1}^{d} x_j^{x_j}}{N^N}\frac{N!}{\prod_{j=1}^{d} x_j!}, \tag{31}$$

which recovers the multinomial density $p(\mathbf{x}) = \exp(-D_\phi(\mathbf{x}, \boldsymbol{\mu}))b_\phi(\mathbf{x})$. Therefore, when the labels are generated from a multinomial distribution, the corresponding Bregman divergence $D_\phi(\mathbf{x}, \boldsymbol{\mu}) = -\sum_{j=1}^{d} x_j \log \mu_j + $ constant recovers the cross-entropy loss commonly used for classification.

## D   LEARNING THE ANCHOR EMBEDDINGS $\mathbf{A}$

Here we provide several other strategies for initializing the anchor embeddings:

- Sparse lasso and variational dropout (Chen et al., 2019). Given the strong performance of sparse lasso and variational dropout as vocabulary selection methods, it would be interesting to use sparse lasso/variational dropout to first select the important task-specific words before jointly learning their representations and their transformations to other words. However, sparse lasso and variational dropout require first training a model to completion unlike frequency and clustering based vocabulary selection methods that can be performed during data preprocessing.
- Coresets involve constructing a reduced data set which can be used as proxy for the full data set, with provable guarantees such that the same algorithm run on the coreset and the full data set gives approximately similar results (Phillips, 2016; Har-Peled & Mazumdar, 2004). Coresets can be approximately computed quickly (Bachem et al., 2017) and can be used to initialize the set of anchors $A$.

In general, there is a trade-off between how quickly we can choose the anchor objects and their performance. Randomly picking anchor objects (which is equivalent to initializing the anchor embeddings with dynamic basis vectors) becomes similar to learning a low-rank factorization of the embedding matrix (Sedov & Yang, 2018), which works well for general cases but can be improved for task-specific applications or with domain knowledge. Stronger vocabulary selection methods like variational dropout and group lasso would perform better but takes significantly longer time to learn. We found that intermediate methods such as frequency, clustering, with WordNet/co-occurrence information works well while ensuring that the preprocessing and training stages are relatively quick.

In Appendix K we provide more results for different initialization strategies including those based on clustering initializations. In general, performance is robust with respect to the choice of $A$ among the ones considered (i.e., random, frequency, and clustering). While frequency and clustering work better, using a set of dynamic basis embeddings still gives strong performance, especially when combined with domain knowledge from WordNet and co-occurrence statistics. This implies that when the user has more information about the discrete objects (e.g., having a good representation space to perform clustering), then the user should do so. However, for a completely new set of discrete objects, simply using low-rank basis embeddings with sparsity also work well.

## E   TRANSFORM: LEARNING A SPARSE $\mathbf{T}$

In addition to a simple sparse linear transformation, we describe some extensions that improve sparsity and expressitivity of the learned representations.

**Reducing redundancy in representations:** To further reduce redundancy in our sparse representations, we perform orthogonal regularization of dynamic basis vectors $\mathbf{A}$ by adding the loss term

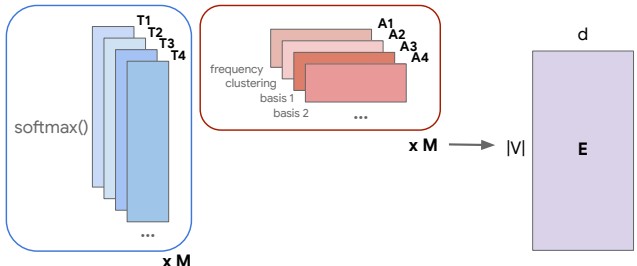

Figure 5: Generalized nonlinear mixture of anchors $\mathbf{A}_1, ..., \mathbf{A}_M$ and transformations $\mathbf{T}_1, ..., \mathbf{T}_M$, $\mathbf{E} = \sum_{m=1}^{M} \mathrm{softmax}(\mathbf{T}_m)\mathbf{A}_m$ (softmax across rows of $\mathbf{T}_m$). Different sparse transformations can be learned for different initializations of anchor embeddings.

$\mathcal{L}(\mathbf{A}) = \sum_{i \neq j} \left| \mathbf{a}_i^\top \mathbf{a}_j \right|$ to the loss function in eq (1). This ensures that different basis vectors $\mathbf{a}_i$ and $\mathbf{a}_j$ are orthogonal instead of being linear combinations of one another which would lead to redundancies across different learnt entries in $\mathbf{T}$.

**Mixture of anchors:** In general, different initialization strategies may bring about different advantages. For example, using a mixture of random basis vectors has been shown to help model multisense embeddings (Athiwaratkun et al., 2018; Nguyen et al., 2017). One can define a set of $M$ anchor embeddings $\mathbf{A}_1, ..., \mathbf{A}_M$ each initialized by different strategies and of possibly different sizes.

**Nonlinear mixture of transformations:** To complement learning multiple sets of anchor embeddings $\mathbf{A}_1, ..., \mathbf{A}_M$, the straightforward extension of the TRANSFORM step would be to learn a separate linear transformation for each anchor embedding and summing the result: $\mathbf{E} = \sum_{m=1}^{M} \mathbf{T}_m \mathbf{A}_m$. However, the expressive power of this linear combination is equivalent to one set of anchor embeddings equal to concatenating $\mathbf{A}_1, ..., \mathbf{A}_M$ and one linear transformation. To truly exhibit the advantage of multiple anchors, we transform and combine them in a nonlinear fashion, e.g., $\mathbf{E} = \sum_{m=1}^{M} \mathrm{softmax}(\mathbf{T}_m)\mathbf{A}_m$ (softmax over the rows of $\mathbf{T}_m$, Figure 5). Different transformations can be learned for different initializations of anchors. This is connected with the multi-head attention mechanism in the Transformer (Vaswani et al., 2017), where $\mathrm{softmax}(\mathbf{T}_m)$ are the softmax-activated (sparse) attention weights and $\mathbf{A}_m$ the values to attend over. The result is an embedding matrix formed via a nonlinear mixture of anchors (each initialized with different strategies) and sparse transformations.

## F   INCORPORATING DOMAIN KNOWLEDGE

ANT also allows incorporating *domain knowledge* about object relationships. Suppose we are given some relationship graph $G = (V, E)$ where each object is a vertex $v \in V$ and an edge $(u, v) \in E$ exists between objects $u$ and $v$ if they are related. Real-world instantiations of such a graph include 1) WordNet (Miller, 1995) or ConceptNet (Liu & Singh, 2004) for semantic relations between words, 2) word co-occurrence matrices (Haralick et al., 1973), and 3) Movie Clustering datasets (Leskovec & Krevl, 2014). From these graphs, we extract related positive pairs $\mathbf{P} = \{(u, v) \in E\}$ and unrelated negative pairs $\mathbf{N} = \{(u, v) \notin E\}$. We incorporate domain information as follows (see Figure 6 for a visual example):

**Positive pairs:** To incorporate a positive pair $(u, v)$, we *do not* enforce sparsity on $\mathbf{T}_{u,v}$. This allows ANT to freely learn the transformation between related objects $u$ and $v$ without being penalized for sparsity. On the other hand, transformations between negative pairs will be sparsely penalized. In other words, before computing the $\ell_1$-penalty, we element-wise multiply $\mathbf{T}$ with a *domain sparsity matrix* $\mathbf{S}(G)$ where $\mathbf{S}(G)_{u,v} = 0$ for $(u, v) \in \mathbf{P}$ (entries not $\ell_1$-penalized) and $\mathbf{S}(G)_{u,v} = 1$ otherwise (entries are $\ell_1$-penalized), resulting in the following modified objective:

$$\min_{\mathbf{T} \geq 0, \; \mathbf{A}, \theta} \sum_i D_\phi(y_i, f_\theta(x_i, \mathbf{TA})) + \lambda_2 \|\mathbf{T} \odot \mathbf{S}(G)\|_1. \tag{32}$$

Since we perform proximal GD, this is equivalent to only soft-thresholding the entries between unrelated objects, i.e., $\mathbf{T} = \max \{(\mathbf{T} - \eta\lambda_2) \odot \mathbf{S}(G) + \mathbf{T} \odot (\mathbf{1} - \mathbf{S}(G)), 0\}$. Note that this strategy is applicable when anchors are selected using the frequency method.

**Negative pairs:** For negative pairs, we add an additional constraint that unrelated pairs should not share entries in their linear combination coefficients of the anchor embeddings. In other words, we

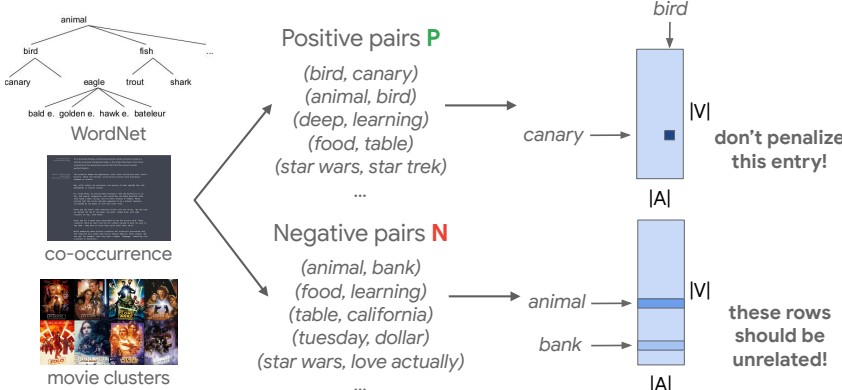

Figure 6: Incorporating domain knowledge into ANT given object relationship graphs (left) with extracted positive pairs **P** and negative pairs **N**. Transformations between negative (unrelated) pairs are sparsely $\ell_1$-penalized while those between positive (related) pairs are not. The linear transformation coefficients $\mathbf{t}_u$ and $\mathbf{t}_v$ of negative pairs $(u, v)$ are also discouraged from sharing similar entries.

add the loss term

$$\mathcal{L}(\mathbf{T}, \mathbf{N}) = \sum_{(u,v) \in \mathbf{N}} |\mathbf{t}_u|^\top |\mathbf{t}_v| \tag{33}$$

to the loss in eq (1), where each inner sum discourages $\mathbf{t}_u$ and $\mathbf{t}_v$ from sharing similar entries. This strategy can used regardless of the way anchors are selected. We acknowledge that there are other ways to incorporate domain knowledge as well into the general ANT framework, and we only serve to give some initial examples of such methods.

## G  NONPARAMETRIC ANCHOR & TRANSFORM

In this section we provide details for our non-parametric extension of ANT. Recall that our full objective function derived via small variance asymptotics is given by:

$$\min_{\substack{\mathbf{T} \geq 0 \\ \mathbf{A}, \theta, K}} \sum_i D_\phi(y_i, f_\theta(x_i; \mathbf{TA})) + \lambda_2 \|\mathbf{T}\|_0 + (\lambda_1 - \lambda_2)K, \tag{34}$$

which suggests a natural objective function in learning representations that minimize the prediction loss $D_\phi(y_i, f_\theta(x_i; \mathbf{TA}))$ while ensuring sparsity of $\mathbf{T}$ as measured by the $\ell_0$-norm and using as few anchors as possible ($K$). Therefore, optimizing eq (5) gives rise to a nonparametric version of ANT, which we call NBANT, that automatically learns the optimal number of anchors. To perform optimization over the number of anchors, our algorithm starts with a small initial number of anchors $K = |A| = 10$ and either adds $\Delta K$ anchors (i.e., adding $\Delta K$ new rows to $\mathbf{A}$ and $\Delta K$ new sparse columns to $\mathbf{T}$) or deletes $\Delta K$ anchors to minimize eq (34) at every epoch depending on the trend of the objective evaluated on the training set. We detail the full Algorithm 2, and highlight the main changes as compared to ANT.

Practically, this algorithm involves the same number of training epochs and batches through each training epoch as the vanilla ANT method. To enable sharing of trained anchors, we change the indices from where $\mathbf{A}$ and $\mathbf{T}$ are read from so that the partially trained removed anchors are still stored in case more anchors need to be added again.

## H  EFFICIENT LEARNING AND INFERENCE

The naive method for learning $\mathbf{E}$ from anchor embeddings $\mathbf{A}$ and the sparse transformations $\mathbf{T}$ still scales linearly with $|V| \times d$. Here we describe some tips on how to perform efficient learning and inference of the anchor embeddings $\mathbf{A}$ and the sparse transformations $\mathbf{T}$:

- Store $\mathbf{T}$ as a sparse matrix by only storing its non-zero entries and indices. From our experiments, we have shown that $\text{nnz}(\mathbf{T}) << |V| \times d$ which makes storage efficient.
- For inference, use sparse matrix multiply as supported in TensorFlow and PyTorch to compute $\mathbf{E} = \mathbf{TA}$ (or its non-linear extensions). This decreases the running time from scaling by $|V| \times d$ to only scaling as a function of $\text{nnz}(\mathbf{T})$. For training, using inbuilt sparse

---

**Algorithm 2** NBANT: Nonparametric Bayesian ANT. Differences from ANT are highlighted in red.

---

**ANCHOR & TRANSFORM:**
1: Anchor: initialize initial $K = |A|$ and corresponding anchor embeddings $\mathbf{A} \in \mathbb{R}^{K \times d}$.
2: Transform: initialize $\mathbf{T} \in \mathbb{R}^{|V| \times K}$ as a *sparse matrix*.
3: **for** each each epoch **do**
4:     **for** each batch $(\mathbf{X}, \mathbf{Y})$ **do**
5:         Compute loss $\mathcal{L} = \sum_i D_\phi(y_i, f_\theta(x_i; \mathbf{TA}))$
6:         $\mathbf{A}, \mathbf{T}, \theta = \text{UPDATE } (\nabla \mathcal{L}, \eta)$.
7:         $\mathbf{T} = \max \{\mathbf{T} - \eta \lambda_2, 0\}$.
8:     **end for**
9:     Compute eq (34) using current value of $K, \mathbf{A}, \mathbf{T}$ on the validation set.
10:     **if** eq (34) is on a decreasing trend **then**
11:         $K = K + \Delta K$, add $\Delta K$ rows to $\mathbf{A}$ and $\Delta K$ (sparse) columns to $\mathbf{T}$.
12:     **else if** eq (34) is on an increasing trend **then**
13:         $K = K - \Delta K$, remove $\Delta K$ rows from $\mathbf{A}$ and $\Delta K$ (sparse) columns from $\mathbf{T}$.
14:     **else**
15:         keep current values of $K, \mathbf{A}, \mathbf{T}$.
16:     **end if**
17: **end for**
18: **return** anchor embeddings $\mathbf{A}$ and transformations $\mathbf{T}$.

---

    representation of most deep learning frameworks like PyTorch or Tensorflow is not optimal, as they do not support changing non-zero locations in sparse matrix and apriori its not easy to find optimal set of non-zero locations.

- During training, instead, implicitly construct $\mathbf{E}$ from its anchors and transformations. In fact, we can do better: instead of constructing the entire $\mathbf{E}$ matrix to embed a single datapoint $\mathbf{x} \in \mathbb{R}^{1 \times |V|}$, we can instead first *index* $\mathbf{x}$ into $\mathbf{T}$, i.e., $\mathbf{xT} \in \mathbb{R}^{1 \times |A|}$ before performing a sparse matrix multiplication with $\mathbf{A}$, i.e., $(\mathbf{xT})\mathbf{A} \in \mathbb{R}^{1 \times d}$. We are essentially taking advantage of the *associative* property of matrix multiplication and the fact that $\mathbf{xT}$ is a simple indexing step and $(\mathbf{xT})\mathbf{A}$ is an effective sparse matrix multiplication. To enable fast row slicing into sparse matrix, we just storing the matrix in adjacency list or CSOO format. (We move away from CSR as adding/deleting a non-zero location is very expensive.) When gradient comes back, only update the corresponding row in $\mathbf{T}$. The gradient will be sparse as well due to the L1-prox operator.

- Above trick solves the problem for tasks where embedding is used only at the input, e.g., classification. For tasks like language model, where embedding is used at output as well one can also use above mentioned trick with speedup techniques like various softmax sampling techniques (Bengio & Senecal, 2008; Mikolov et al., 2013) or noise-contrastive estimation (Gutmann & Hyvarinen, 2010; Mnih & Teh, 2012), which will be anyway used for large vocabulary sizes. To elaborate, consider the case of sampled softmax (Bengio & Senecal, 2008). We normally generate the negative sample indices, and then we can first *index* into $\mathbf{T}$ using the true and negative indices before performing sparse matrix multiplication with $\mathbf{A}$. This way we do not have to instantiate entire $\mathbf{E}$ by expensive matrix multiplication.

- When training is completed, only store the non-zero entries of $\mathbf{T}$ or store $\mathbf{T}$ as a sparse matrix to reconstruct $\mathbf{E}$ for inference.

- To save time when initializing the anchor embeddings and incorporating domain knowledge, precompute the necessary statistics such as frequency statistics, co-occurrence statistics, and object relation statistics. We use a small context size of 10 to measure co-occurrence of two words to save time. When using WordNet to discover word relations, we only search for immediate relations between words instead of propagating relations across multiple steps (although this could further improve performance).

- In order to incorporate domain knowledge in the sparsity structure, we again store $\mathbf{1} - \mathbf{S}(G)$ using sparse matrices. Recall that $\mathbf{S}(G)$ has an entry equal to 1 for entries representing unrelated objects that should be $\ell_1$-penalized, which makes $\mathbf{S}(G)$ quite dense since most anchor and non-anchor objects are unrelated. Hence we store $\mathbf{1} - \mathbf{S}(G)$ instead which consists few non-zero entries only at (non-anchor, anchor) entries for related objects. Element-wise multiplications are also replaced by sparse element-wise multiplications when computing $\mathbf{T} \odot \mathbf{S}(G)$ and $\mathbf{T} \odot (\mathbf{1} - \mathbf{S}(G))$.

Table 6: Table of hyperparameters for text classification experiments on AG-News, DBPedia, Sogou-News, and Yelp-review datasets. All text classification experiments use the same base CNN model with the exception of different output dimensions (classes in the dataset): 4 for AG-News, 14 for DBPedia, 5 for Sogou-News, and 5 for Yelp-review.

| Model | Parameter | Value |
|-------|-----------|-------|
|       | Embedding dim | 256 |
|       | Filter sizes | $[3, 4, 5]$ |
|       | Num filters | 100 |
|       | Filter strides | $[1, 1]$ |
|       | Filter padding | valid |
|       | Pooling strides | $[1, 1]$ |
|       | Pooling padding | valid |
|       | Loss | cross entropy |
| CNN   | Dropout | 0.5 |
|       | Batch size | 256 |
|       | Max seq length | 100 |
|       | Num epochs | 200 |
|       | Activation | ReLU |
|       | Optimizer | Adam |
|       | Learning rate | $5 \times 10^{-3}$ |
|       | Learning rate decay | $1 \times 10^{-5}$ |
|       | Start decay | 40 |

- Finally, even if we want to use our ANT framework with full softmax in language model, it is possible without blowing up memory requirements. In particular, let $\mathbf{g} \in \mathbf{R}^{1 \times |V|}$ be the incoming gradient from cross-entropy loss and $\mathbf{h} \in \mathbf{R}^{d \times 1}$ be the vector coming from layers below, like LSTM. The gradient update is then

$$\mathbf{T} \leftarrow \text{PROX}_{\eta\lambda}(\mathbf{T} - \eta\mathbf{g}(\mathbf{Ah})^T) \tag{35}$$

The main issue is computing the huge $|V| \times |A|$ outer product as an intermediate step which will be dense. However, note that incoming gradient $\mathbf{g}$ is basically a softmax minus an offset corresponding to correct label. This should only have large values for a small set of words and small for others. If we carefully apply the $\ell_1$-prox operator earlier, which is nothing but a soft-thresholding, we can make this incoming gradient sparse very sparse. Thus we need to only calculate a much smaller sized outer product and touch a small number of rows in $\mathbb{T}$. Thus, making the approach feasible.

# I   GENERALITY OF ANT

We show that under certain structural assumptions on the anchor embeddings and transformation matrices, ANT reduces to the following task-specific methods for learning sparse representations: 1) Frequency (Chen et al., 2016b), TF-IDF, Group Lasso (Wen et al., 2016), and variational dropout (Chen et al., 2019) based vocabulary selection, 2) Low-rank factorization (Grachev et al., 2019), and 3) Compositional code learning (Shu & Nakayama, 2018; Chen et al., 2018). Hence, ANT is general and unifies some of the work on sparse representation learning done independently in different research areas.

**Frequency-based vocabulary selection** (Luong et al., 2015; Chen et al., 2016b): Initialize $A$ with the $|A|$ most frequent objects and set $\mathbf{T}_{a,a} = 1$ for all $a \in A$, $\mathbf{T} = 0$ otherwise. Then $\mathbf{E} = \mathbf{TA}$ consists of embeddings of the $|A|$ most frequent objects with zero embeddings for all others. During training, gradients are used to update $\mathbf{A}$ but not $\mathbf{T}$ (i.e., only embeddings for frequent objects are learned). By changing the selection of $A$, ANT also reduces to other vocabulary selection methods such as TF-IDF (Ramos, 1999), Group Lasso (Wen et al., 2016), and variational dropout (Chen et al., 2019)

**Low-rank factorization** (Acharya et al., 2019; Markovsky, 2011; Grachev et al., 2019): Initialize $A$ by a mixture of random basis embeddings (just 1 anchor per set) $\mathbf{A}_1, ..., \mathbf{A}_M \in \mathbb{R}^{1 \times d}$ and do not enforce any sparsity on the transformations $\mathbf{T}_1, ..., \mathbf{T}_M \in \mathbb{R}^{|V| \times 1}$. If we further restrict ourselves to only linear combinations $\mathbf{E} = \sum_{m=1}^{M} \mathbf{T}_m \mathbf{A}_m$, this is equivalent to implicitly learning the $M$ low rank factors $\mathbf{a}_1, ..., \mathbf{a}_M, \mathbf{t}_1, ..., \mathbf{t}_M$ that reconstruct embedding matrices of rank at most $M$.

Table 7: Table of hyperparameters for language modeling experiments using LSTM on PTB dataset.

| Model | Parameter | Value |
|---|---|---|
| | Embedding dim | 200 |
| | Num hidden layers | 2 |
| | Hidden layer size | 200 |
| | Output dim | 10, 000 |
| | Loss | cross entropy |
| | Dropout | 0.4 |
| | Word embedding dropout | 0.1 |
| | Input embedding dropout | 0.4 |
| | LSTM layers dropout | 0.25 |
| | Weight dropout | 0.5 |
| LSTM | Weight decay | $1.2 \times 10^{-6}$ |
| | Activation regularization | 2.0 |
| | Temporal activation regularization | 1.0 |
| | Batchsize | 20 |
| | Max seq length | 70 |
| | Num epochs | 500 |
| | Activation | ReLU |
| | Optimizer | SGD |
| | Learning rate | 30 |
| | Gradient clip | 0.25 |
| | Learning rate decay | $1 \times 10^{-5}$ |
| | Start decay | 40 |

**Compositional code learning** (Shu & Nakayama, 2018; Chen et al., 2018): Initialize $A$ by a mixture of random basis embeddings $\mathbf{A}_1, ..., \mathbf{A}_M$, initialize transformations $\mathbf{T}_1, ..., \mathbf{T}_M$, and apply a linear combination $\mathbf{E} = \sum_{m=1}^{M} \mathbf{T}_m \mathbf{A}_m$. For sparsity regularization, set row $i$ of $\mathbf{S}(G)_{mi}$ as a reverse one-hot vector with entry $d_{mi} = 0$ and all else 1. In other words, index $d_{mi}$ of row row $\mathbf{T}_{mi}$ is not regularized, and all other entries are $\ell_1$-regularized with extremely high $\lambda_2$ such that row $\mathbf{T}_{mi}$ essentially becomes an one-hot vector with dimension $d_{mi} = 1$. This results in learning a *codebook* where each object in $V$ is mapped to *only one* anchor in each mixture.

Therefore, ANT encompasses several popular methods for learning sparse representations, and gives further additional flexibility in defining various initialization strategies, applying nonlinear mixtures of transformations, and incorporating domain knowledge via object relationships.

## J  EXPERIMENTAL DETAILS

Here we provide more details for our experiments including hyperparameters used, design decisions, and comparison with baseline methods. We also include the anonymized code in the supplementary material.

### J.1  TEXT CLASSIFICATION

**Base CNN model:** For all text classification experiments, the base model is a CNN (Lecun et al., 1998) with layers of 2D convolutions and 2D max pooling, before a dense layer to the output softmax. The code was adapted from `https://github.com/wenhuchen/Variational-Vocabulary-Selection` and the architecture hyperparameters are provided in Table 6. The only differences are the output dimensions which is 4 for AG-News, 14 for DBPedia, 5 for Sogou-News, and 5 for Yelp-review.

**Anchor:** We experiment with dynamic, frequency, and clustering initialization strategies. The number of anchors $|A|$ is a hyperparameter that is selected using the validation set. The range of $|A|$ is in $\{10, 20, 50, 80, 100, 500, 1, 000\}$. Smaller values of $|A|$ allows us to control for fewer anchors and smaller transformation matrix $\mathbf{T}$ at the expense of performance.

**Transformation:** We experiment with sparse linear transformations for $\mathbf{T}$. $\lambda_2$ is a hyperparameter that is selected using the validation set. Larger values of $\lambda_2$ allows us to control for more sparse entries in $\mathbf{T}$ at the expense of performance. For experiments on dynamic mixtures, we use a softmax-based nonlinear combination $\mathbf{E} = \sum_{m=1}^{M} \text{softmax}(\mathbf{T}_m) \mathbf{A}_m$ where softmax is performed over the rows of

Table 8: Table of hyperparameters for language modeling experiments using AWD-LSTM on PTB dataset.

| Model | Parameter | Value |
|---|---|---|
| | Embedding dim | 400 |
| | Num hidden layers | 3 |
| | Hidden layer size | $1,150$ |
| | Output dim | $10,000$ |
| | Loss | cross entropy |
| | Dropout | 0.4 |
| | Word embedding dropout | 0.1 |
| | Input embedding dropout | 0.4 |
| | LSTM layers dropout | 0.25 |
| | Weight dropout | 0.5 |
| AWD-LSTM | Weight decay | $1.2 \times 10^{-6}$ |
| | Activation regularization | 2.0 |
| | Temporal activation regularization | 1.0 |
| | Batchsize | 20 |
| | Max seq length | 70 |
| | Num epochs | 500 |
| | Activation | ReLU |
| | Optimizer | SGD |
| | Learning rate | 30 |
| | Gradient clip | 0.25 |
| | Learning rate decay | $1 \times 10^{-5}$ |
| | Start decay | 40 |

$\mathbf{T}_m$. Note that applying a softmax activation to the rows of $\mathbf{T}_m$ makes all entries dense so during training, we store $\mathbf{T}_m$ as sparse matrices (which is efficient since $\mathbf{T}_m$ has few non-zero entries) and *implicitly* reconstruct $\mathbf{E}$.

**Domain knowledge:** When incorporating domain knowledge in ANT, we use both WordNet and co-occurrence statistics. For WordNet, we use the public WordNet interface provided by NLTK `http://www.nltk.org/howto/wordnet.html`. For each word we search for its immediate related words among its hypernyms, hyponyms, synonyms, and antonyms. This defines the relationship graph. For co-occurrence statistics, we define a co-occurrence context size of 10 on the training data. Two words are defined to be related if they co-occur within this context size.

**A note on baselines:** Note that the reported results on SPARSEVD and SPARSEVD-VOC (Chirkova et al., 2018) have a different embedding size: 300 instead of 256. This is because they use pre-trained word2vec or GloVe embeddings to initialize their model before compression is performed.

### J.2 LANGUAGE MODELING ON PTB

**Base LSTM model:** Our base model is a 2 layer LSTM with an embedding size of 200 and hidden layer size of 200. The code was adapted from `https://github.com/salesforce/awd-lstm-lm` and the full table of hyperparameters is provided in Table 7.

**Base AWD-LSTM model:** In addition to experiments on an vanilla LSTM model as presented in the main text, we also performed experiments using a 3 layer AWD-LSTM with an embedding size of 400 and hidden layer size of $1,150$. The full hyperparameters used can be found in Table 8.

**Anchor:** We experiment with dynamic, frequency, and clustering initialization strategies. The number of anchors $|A|$ is a hyperparameter that is selected using the validation set. The range of $|A|$ is in $\{10, 20, 50, 80, 100, 500, 1,000\}$. Smaller values of $|A|$ allows us to control for fewer anchors and smaller transformation matrix $\mathbf{T}$ at the expense of performance.

**Domain knowledge:** When incorporating domain knowledge in ANT, we use both WordNet and co-occurrence statistics. For WordNet, we use the public WordNet interface provided by NLTK `http://www.nltk.org/howto/wordnet.html`. For each word we search for its immediate related words among its hypernyms, hyponyms, synonyms, and antonyms. This defines the relationship graph. For co-occurrence statistics, we define a co-occurrence context size of 10 on the training data. Two words are defined to be related if they co-occur within this context size.

Table 9: Table of hyperparameters for language modeling experiments using AWD-LSTM on WikiText-103 dataset.

| Model | Parameter | Value |
|---|---|---|
|  | Embedding dim | 400 |
|  | Num hidden layers | 4 |
|  | Hidden layer size | $2,500$ |
|  | Output dim | $267,735$ |
|  | Loss | cross entropy |
|  | Dropout | 0.1 |
|  | Word embedding dropout | 0.0 |
|  | Input embedding dropout | 0.1 |
|  | LSTM layers dropout | 0.1 |
|  | Weight dropout | 0.0 |
| AWD-LSTM | Weight decay | 0.0 |
|  | Activation regularization | 0.0 |
|  | Temporal activation regularization | 0.0 |
|  | Batchsize | 32 |
|  | Max seq length | 140 |
|  | Num epochs | 14 |
|  | Activation | ReLU |
|  | Optimizer | SGD |
|  | Learning rate | 30 |
|  | Gradient clip | 0.25 |
|  | Learning rate decay | $1 \times 10^{-5}$ |
|  | Start decay | 40 |

**A note on baselines:** We also used some of the baseline results as presented in Grachev et al. (2019). Their presented results differ from our computations in two aspects: they include the LSTM parameters on top of the embedding parameters, and they also count the embedding parameters twice since they do not perform weight tying (Press & Wolf, 2017) (see equation (6) of Grachev et al. (2019)). To account for this, the results of SPARSEVD and SPARSEVD-VOC (Chirkova et al., 2018), as well as the results of various LR and TT low rank compression methods (Grachev et al., 2019) were modified by subtracting off the LSTM parameters ($200 \times 200 \times 16$). This is derived since each of the 8 weight matrices $W_{i,f,o,c}, U_{i,f,o,c}$ in an LSTM layer is of size $200 \times 200$, and there are a 2 LSTM layers. We then divide by two to account for weight tying. In the main text, we compared with the *strongest* baselines as reported in Grachev et al. (2019): these were the methods that performed low rank decomposition on both the input embedding ($|V| \times d$), output embedding ($d \times |V|$), and intermediate hidden layers of the model. For full results, please refer to Grachev et al. (2019).

Note that the reported results on SPARSEVD and SPARSEVD-VOC (Chirkova et al., 2018) have a different embedding size and hidden layer size of 256 instead of 200, although these numbers are close enough for fair comparison. In our experiments we additionally implemented an LSTM with an embedding size of 256 and hidden layer size of 256 so that we can directly compare with their reported numbers.

For baselines that perform post-processing compression of the embedding matrix, POST-SPARSE HASH (post-processing using sparse hashing) (Guo et al., 2017) and POST-SPARSE HASH+$k$-SVD (improving sparse hashing using $k$-SVD) (Guo et al., 2017; Awasthi & Vijayaraghavan, 2018), we choose two settings: the first using $500$ anchors and $10$ nearest neighbors to these anchor points, and the second using $1,000$ anchors and $20$ nearest neighbors. The first model uses $500 \times d + |V| \times 10$ non-zero embedding parameters while the second model uses $1,000 \times d + |V| \times 20$ parameters. For AWD-LSTM on PTB, this is equivalent to 0.3M and 0.6M embedding parameters respectively which is comparable to the number of non-zero parameters used by our method.

## J.3 LANGUAGE MODELING ON WIKITEXT-103

**Base AWD-LSTM model:** Our base model is a $4$ layer AWD-LSTM with an embedding size of $400$ and hidden layer size of $2,500$. The code was adapted from `https://github.com/salesforce/awd-lstm-lm` and the hyperparameters used can be found in Table 9.

Table 10: Table of hyperparameters for movie recommendation experiments on Movielens 1M (top) and Movielens 25M (bottom). Initial $K$ and $\Delta K$ are used for NBANT experiments.

| Model | Parameter | Value |
|-------|-----------|-------|
|       | Embedding dim | 16 |
|       | Initial $K$ | 10 |
|       | $\Delta K$ | 1 |
|       | Loss | mse |
|       | Batch size | 32 |
| MF    | Num epochs | 50 |
|       | Optimizer | Yogi |
|       | Learning rate | 0.01 |
|       | Learning rate decay | 0.5 |
|       | Decay step size | $100,000$ |

| Model | Parameter | Value |
|-------|-----------|-------|
|       | Embedding dim | 16 |
|       | Initial $K$ | 20 |
|       | $\Delta K$ | 2 |
|       | Loss | mse |
|       | Batch size | $1,024$ |
| MF    | Num epochs | 50 |
|       | Optimizer | Yogi |
|       | Learning rate | 0.01 |
|       | Learning rate decay | 0.5 |
|       | Decay step size | $200,000$ |

**A note on baselines:** While Baevski & Auli (2019) adapt embedding dimensions according to word frequencies, their goal is not to compress embedding parameters and they use 44.9M (dense) parameters in their adaptive embedding layer, while we use only 2M. Their embedding parameters are calculated by their reported bucket sizes and embedding sizes (three bands of size 20K ($d = 1024$), 40K ($d = 256$) and 200K ($d = 64$)). Their perplexity results are also obtained using a Transformer model with 250M params while our AWD-LSTM model uses 130M params.

For the HASH EMBED baseline that retains the frequent $k$ words and hashes the remaining words into $1,000$ OOV buckets (Svenstrup et al., 2017), We vary $k \in \{1 \times 10^5, 5 \times 10^4, 1 \times 10^4\}$ to obtain results across various parameter settings.

### J.4 MOVIE RECOMMENDATION ON MOVIELENS

**Base MF model:** We show the hyperparamters used for the MF model in Table 10. We use the Yogi optimizer (Zaheer et al., 2018) to learn the parameters.

**ANT and NBANT**: We build ANT on top of the MF model while keeping the base hyperparamters constant. For ANT, we apply compression to both movie and user embedding matrices individually. NBANT involves defining the starting value of $K = |A|$, and a $\Delta K$ value which determines the rate of increase or decrease in $K$. For Movielens 25M we use a larger initial $|A|$ and $\Delta K$ since it is a larger dataset and also takes longer to train, so we wanted the increase and decrease in anchors to be faster (see Table 10). Beyond this initial setting, we found that performance is robust with respect to the initial value of $K$ and $\Delta K$, so we did not tune these parameters. In practice, we tie the updates of the number of user anchors and movie anchors instead of optimizing over both independently. Therefore, we start with the same number of initial user and movie anchors before incrementing or decrementing them by the same $\Delta K$ at the same time. We found that this simplification did not affect performance and NBANT was still able to find an optimal number of anchors for a good trade-off between performance and compression.

## K MORE RESULTS

In the following sections, we provide additional results on learning sparse representations of discrete objects using ANT.

Table 11: More text classification results on (from top to bottom) AG-News, DBPedia, Sogou-News, and Yelp-review. Domain knowledge is derived from WordNet and co-occurrence statistics. Our approach with different initializations and domain knowledge achieves within $1\%$ accuracy with $21\times$ fewer parameters on DBPedia, within $1\%$ accuracy with $10\times$ fewer parameters on Sogou-News, and within $2\%$ accuracy with $22\times$ fewer parameters on Yelp-review. Acc: accuracy, # Emb: # (non-zero) embedding parameters.

| Methods on **AG-News** | $\|A\|$ | Init $A$ | Sparse $\mathbf{T}$ | $\mathbf{T} \geq 0$ | Domain | Acc (%) | # Emb (M) |
|---|---|---|---|---|---|---|---|
| CNN (Zhang et al., 2015) | $61,673$ | All | ✗ | ✗ | ✗ | 91.6 | 15.87 |
| FREQUENCY (Chen et al., 2019) | $5,000$ | Frequency | ✗ | ✗ | ✗ | 91.0 | 1.28 |
| TF-IDF (Chen et al., 2019) | $5,000$ | TF-IDF | ✗ | ✗ | ✗ | 91.0 | 1.28 |
| GL (Chen et al., 2019) | $4,000$ | Group lasso | ✗ | ✗ | ✗ | 91.0 | 1.02 |
| VVD (Chen et al., 2019) | $3,000$ | Var dropout | ✗ | ✗ | ✗ | 91.0 | 0.77 |
| SPARSEVD (Chirkova et al., 2018) | $5,700$ | Mult weights | ✗ | ✗ | ✗ | 88.8 | 1.72 |
| SPARSEVD-VOC (Chirkova et al., 2018) | $2,400$ | Mult weights | ✗ | ✗ | ✗ | 89.2 | 0.73 |
| SPARSE CODE (Chen et al., 2016b) | $100$ | Frequency | ✓ | ✗ | ✗ | 89.5 | 2.03 |
| ANT | 50 | Frequency | ✓ | ✓ | ✗ | 89.5 | 1.01 |
| | 10 | Frequency | ✓ | ✓ | ✓ | **91.0** | **0.40** |
| | 10 | Random | ✓ | ✓ | ✓ | 90.5 | 0.40 |
| | 5 | Random mixture | ✓ | ✓ | ✓ | 90.5 | 0.70 |

| Methods on **DBPedia** | $\|A\|$ | Init $A$ | Sparse $\mathbf{T}$ | $\mathbf{T} \geq 0$ | Domain | Acc (%) | # Emb (M) |
|---|---|---|---|---|---|---|---|
| CNN (Zhang et al., 2015) | $563,355$ | All | ✗ | ✗ | ✗ | 98.3 | 144.0 |
| SPARSE CODE (Chen et al., 2016b) | $100$ | Frequency | ✓ | ✗ | ✗ | 96.7 | 39.0 |
| ANT | 80 | Cluster | ✓ | ✓ | ✗ | 98.1 | 30.0 |
| | 100 | Random | ✓ | ✓ | ✗ | 98.2 | 28.0 |
| | 50 | Frequency | ✓ | ✓ | ✓ | 97.3 | 18.0 |
| | 20 | Frequency | ✓ | ✓ | ✓ | **97.2** | **7.0** |

| Methods on **Sogou-News** | $\|A\|$ | Init $A$ | Sparse $\mathbf{T}$ | $\mathbf{T} \geq 0$ | Domain | Acc (%) | # Emb (M) |
|---|---|---|---|---|---|---|---|
| CNN (Zhang et al., 2015) | $254,495$ | All | ✗ | ✗ | ✗ | 94.0 | 65.0 |
| SPARSE CODE (Chen et al., 2016b) | $100$ | Frequency | ✓ | ✗ | ✗ | 92.0 | 6.0 |
| ANT | 50 | Cluster | ✓ | ✓ | ✗ | **93.0** | **5.0** |
| | 80 | Cluster | ✓ | ✓ | ✗ | 93.1 | 9.0 |
| | 100 | Random | ✓ | ✓ | ✗ | 93.2 | 6.0 |
| | 50 | Frequency | ✓ | ✓ | ✓ | 92.0 | **5.0** |

| Methods on **Yelp-review** | $\|A\|$ | Init $A$ | Sparse $\mathbf{T}$ | $\mathbf{T} \geq 0$ | Domain | Acc (%) | # Emb (M) |
|---|---|---|---|---|---|---|---|
| CNN (Zhang et al., 2015) | $252,712$ | All | ✗ | ✗ | ✗ | 56.2 | 65.0 |
| SPARSE CODE (Chen et al., 2016b) | $100$ | Frequency | ✓ | ✗ | ✗ | 54.0 | 14.0 |
| ANT | 80 | Cluster | ✓ | ✓ | ✗ | 56.2 | 8.0 |
| | 50 | Random | ✓ | ✓ | ✗ | 56.0 | 6.0 |
| | 50 | Frequency | ✓ | ✓ | ✓ | **54.7** | **3.0** |

## K.1 TEXT CLASSIFICATION

**Extra results:** We report additional text classification results on AG-News, DBPedia, Sogou-News, and Yelp-review in Table 11. For AG-News, using a mixture of anchors and transformations also achieves stronger performance than the baselines using 5 anchors per mixture, although the larger number of transformations leads to an increase in parameters. Our approach with different initializations and domain knowledge achieves within $1\%$ accuracy with $21\times$ fewer parameters on DBPedia, within $1\%$ accuracy with $10\times$ fewer parameters on Sogou-News, and within $2\%$ accuracy with $22\times$ fewer parameters on Yelp-review.

**Different initialization strategies:** Here we also presented results across different initialization strategies and find that while those based on frequency and clustering work better, using a set of dynamic basis embeddings still gives strong performance, especially when combined with domain knowledge from WordNet and co-occurrence statistics. This implies that when the user has more information about the discrete objects (e.g., having a good representation space to perform clustering),

Table 12: Language modeling using LSTM (top) and AWD-LSTM (bottom) on PTB. We outperform the existing vocabulary selection, low-rank, tensor-train, and post-compression (hashing) baselines. 200/256 represents the embedding dimension. Incorporating domain knowledge further reduces parameters. Ppl: perplexity, # Emb: number of (non-zero) embedding parameters.

| Method | $|A|$ | Init $A$ | Sparse $\mathbf{T}$ | $\mathbf{T} \geq 0$ | Domain | Ppl | # Emb (M) |
|---|---|---|---|---|---|---|---|
| LSTM 200 (Grachev et al., 2019) | 10,000 | All | ✗ | ✗ | ✗ | 77.1 | 2.00 |
| LSTM 256 (Chirkova et al., 2018) | 10,000 | All | ✗ | ✗ | ✗ | 70.3 | 2.56 |
| LR LSTM 200 (Grachev et al., 2019) | 10,000 | All | ✗ | ✗ | ✗ | 112.1 | 1.26 |
| TT LSTM 200 (Grachev et al., 2019) | 10,000 | All | ✗ | ✗ | ✗ | 116.6 | 1.16 |
| SPARSEVD 256 (Chirkova et al., 2018) | 9,985 | Mult weights | ✗ | ✗ | ✗ | 109.2 | 1.34 |
| SPARSEVD-VOC 256 (Chirkova et al., 2018) | 4,353 | Mult weights | ✗ | ✗ | ✗ | 120.2 | 0.52 |
| ANT 200 | 2,000 | Random | ✓ | ✓ | ✗ | **77.7** | 0.65 |
|  | 1,000 | Random | ✓ | ✓ | ✗ | 79.4 | 0.41 |
|  | 500 | Random | ✓ | ✓ | ✗ | 84.5 | 0.27 |
|  | 100 | Random | ✓ | ✓ | ✗ | 106.6 | **0.05** |
| ANT 256 | 2,000 | Random | ✓ | ✓ | ✗ | **71.5** | 0.78 |
|  | 1,000 | Random | ✓ | ✓ | ✗ | 73.1 | 0.49 |
|  | 500 | Random | ✓ | ✓ | ✗ | 77.2 | 0.31 |
|  | 100 | Random | ✓ | ✓ | ✗ | 96.5 | **0.05** |

| Method | $|A|$ | Init $A$ | Sparse $\mathbf{T}$ | $\mathbf{T} \geq 0$ | Domain | Ppl | # Emb (M) |
|---|---|---|---|---|---|---|---|
| AWD-LSTM (Merity et al., 2018) | 10,000 | All | ✗ | ✗ | ✗ | 59.0 | 4.00 |
| POST-SPARSE HASH (Guo et al., 2017) | 1,000 | Post Processing | ✓ | ✗ | ✗ | 118.8 | 0.60 |
| POST-SPARSE HASH (Guo et al., 2017) | 500 | Post Processing | ✓ | ✗ | ✗ | 166.8 | 0.30 |
| POST-SPARSE HASH+$k$-SVD | 1,000 | Post Processing | ✓ | ✗ | ✗ | 78.0 | 0.60 |
| POST-SPARSE HASH+$k$-SVD | 500 | Post Processing | ✓ | ✗ | ✗ | 103.5 | 0.30 |
| ANT | 1,000 | Random | ✓ | ✓ | ✗ | 72.0 | 0.44 |
|  | 500 | Random | ✓ | ✓ | ✗ | 74.0 | 0.26 |
|  | 1,000 | Frequency | ✓ | ✓ | ✗ | 77.0 | 0.45 |
|  | 100 | Frequency | ✓ | ✓ | ✓ | **70.0** | **0.05** |

Table 13: Language modeling results on WikiText-103. We reach within 3 points perplexity with ~ 16× reduction and within 13 points perplexity with ~ 80× reduction, outperforming the frequency (HASH EMBED) and post-processing hashing (SPARSE HASH) baselines.

| Method | $|A|$ | Init $A$ | Sparse $\mathbf{T}$ | $\mathbf{T} \geq 0$ | Domain | Ppl | # Emb (M) |
|---|---|---|---|---|---|---|---|
| AWD-LSTM (Merity et al., 2018) | 267,735 | All | ✗ | ✗ | ✗ | 35.2 | 106.8 |
| HASH EMBED (Svenstrup et al., 2017) | 100,000 | Frequency | ✗ | ✗ | ✗ | 40.6 | 40.4 |
| HASH EMBED (Svenstrup et al., 2017) | 50,000 | Frequency | ✗ | ✗ | ✗ | 52.5 | 20.4 |
| HASH EMBED (Svenstrup et al., 2017) | 10,000 | Frequency | ✗ | ✗ | ✗ | 70.2 | 4.4 |
| POST-SPARSE HASH (Guo et al., 2017) | 1,000 | Post Processing | ✓ | ✗ | ✗ | 764.7 | 5.7 |
| POST-SPARSE HASH (Guo et al., 2017) | 500 | Post Processing | ✓ | ✗ | ✗ | 926.8 | 2.9 |
| POST-SPARSE HASH+$k$-SVD | 1,000 | Post Processing | ✓ | ✗ | ✗ | 73.7 | 5.7 |
| POST-SPARSE HASH+$k$-SVD | 500 | Post Processing | ✓ | ✗ | ✗ | 148.3 | 2.9 |
| ANT | 1,000 | Random ($\lambda_2 = 1 \times 10^{-6}$) | ✓ | ✓ | ✗ | **38.4** | 6.5 |
|  | 1,000 | Random ($\lambda_2 = 1 \times 10^{-5}$) | ✓ | ✓ | ✗ | 39.7 | 3.1 |
|  | 500 | Random ($\lambda_2 = 1 \times 10^{-6}$) | ✓ | ✓ | ✗ | 48.8 | 1.4 |
|  | 500 | Random ($\lambda_2 = 1 \times 10^{-5}$) | ✓ | ✓ | ✗ | 54.2 | **0.4** |

then the user should do so. However, for a completely new set of discrete objects, simply using low-rank basis embeddings with sparsity also work well.

**Incorporating domain knowledge:** We find that from WordNet and co-occurrence helps to further reduce the total embedding parameters while maintaining task performance.

### K.2 LANGUAGE MODELING

**Extra results on PTB:** We report additional language modeling results using AWD-LSTM on PTB in Table 12. ANT with 1,000 dynamic basis vectors is able to compress the embedding parameters by 10× while achieving 72.0 test perplexity. By incorporating domain knowledge, we further compress

Table 14: On Movielens 1M, ANT outperforms MF and mixed dimensional embeddings. NBANT automatically tunes $|A|$ (*denotes $|A|$ discovered by NBANT) to achieve a balance between performance and compression.

| Method | user $|A|$ | item $|A|$ | Init $A$ | Sparse $\mathbf{T}$ | $\mathbf{T} \geq 0$ | MSE | # Emb (K) |
|---|---|---|---|---|---|---|---|
| MF | 6K | 3.7K | All | ✗ | ✗ | 0.771 | 155.2 |
| MixDim | 6K | 3.7K | All ($d = 16, \alpha = 0.4, k = 8$) | ✗ | ✗ | 1.113 | 66.6 |
| | 6K | 3.7K | All ($d = 16, \alpha = 0.4, k = 16$) | ✗ | ✗ | 1.084 | 66.2 |
| | 6K | 3.7K | All ($d = 16, \alpha = 0.6, k = 8$) | ✗ | ✗ | 1.098 | 47.0 |
| | 6K | 3.7K | All ($d = 16, \alpha = 0.6, k = 16$) | ✗ | ✗ | 1.073 | 42.9 |
| | 6K | 3.7K | All ($d = 32, \alpha = 0.6, k = 8$) | ✗ | ✗ | 1.163 | 94.0 |
| | 6K | 3.7K | All ($d = 32, \alpha = 0.6, k = 16$) | ✗ | ✗ | 1.130 | 84.6 |
| ANT | 120 | 8 | Random ($\lambda_2 = 1 \times 10^{-4}$) | ✓ | ✓ | **0.772** | 59.4 |
| | 15 | 15 | Random ($\lambda_2 = 1 \times 10^{-4}$) | ✓ | ✓ | 0.786 | 29.2 |
| | 5 | 10 | Random ($\lambda_2 = 1 \times 10^{-4}$) | ✓ | ✓ | 0.836 | 10.8 |
| NBANT | Auto → 11* | Auto → 11* | Random ($\lambda_1 = 0.01, \lambda_2 = 1 \times 10^{-4}$) | ✓ | ✓ | 0.795 | 27.7 |
| | Auto → 11* | Auto → 11* | Random ($\lambda_1 = 0.01, \lambda_2 = 2 \times 10^{-4}$) | ✓ | ✓ | 0.837 | **10.6** |

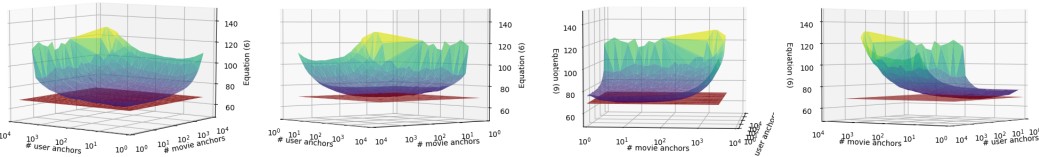

Figure 7: A 3D version of the 2D plot shown in Figure 3 where we simplified the plot by only showing grid points with an equal number of user and movie anchors for Movielens 25M. Here we provide several orientation views of the full 3D plot across user anchors ($x$-axis), movie anchors ($y$-axis), and objective value of eq (34) ($z$-axis). NBANT reaches the objective value indicated by the shaded red plane which is close to the optimal objective value as computed over all grid search experiments for user and movie anchors. Therefore, NBANT can efficiently optimize $|A|$ to achieve a balance between performance and compression, eventually reaching a good value of eq (5) from just one run.

the embedding parameters by *another* 10× and achieve 70.0 test perplexity, which results in 100× total compression as compared to the baseline. We also perform more controlled experiments with different embedding dimension sizes 200 and 250 where we also outperform the baselines.

**Extra results on WikiText-103:** We also report full results using AWD-LSTM on WikiText-103 in Table 13, where we reach within 3 points perplexity with ~ 16× reduction and within 13 points perplexity with ~ 80× reduction, outperforming the frequency (HASH EMBED) and post-processing hashing (SPARSE HASH) baselines.

### K.3    Movie Recommendation

**Extra results on MovieLens 1M:** We also report results on MovieLens 1M in Table 14, where we also observe improvements in accuracy and compression. We also run NBANT on Movielens 1M. Despite the small size of the dataset, NBANT is able to optimize for $|A|$ quickly and achieve a good trade-off between performance and compression.

**Extra results on MovieLens 25M:** Finally, we provide a 3D version of the 2D plot shown in Figure 3 where we simplified the plot by showing grid points with an equal number of user and movie anchors. We provide several orientation views of the full 3D plot across user anchors ($x$-axis), movie anchors ($y$-axis), and objective value of eq (34) ($z$-axis) in Figure 7. NBANT reaches the objective value indicated by the shaded red plane which is close to the optimal objective value as computed over all grid search experiments for user and movie anchors. Therefore, NBANT can efficiently optimize $|A|$ and reach a good value of eq (5) from just one run.

**Online NBANT:** Since NBANT automatically grows/contracts $|A|$ during training, we can further extend NBANT to an online version that sees a stream of batches without revisiting previous ones Bryant & Sudderth (2012). This further enables NBANT to scale to large datasets that cannot all fit in memory. We treat each batch as a new set of data coming in and train on that batch until convergence, modify $|A|$ as in Algorithm 2, before moving onto the next batch. In this significantly more challenging online setting, NBANT is still able to learn well and achieve a MSE of 0.875 with 1.25M non zero parameters.

From Figure 8, we found that initially $|A|$ grew steadily from 10 up to 26 as online batches were seen. As even more batches were seen, the number of clusters decreased steadily from 26 to 10, and oscillated between 8 and 10. This means initially some connections/groupings between online batches were not clear, but with more data, natural clusters merged together. Interestingly this online version of NBANT settled on a similar range of final user (8) and item (8) anchors as compared to the non-online version (see Table 3), which confirms the robustness of NBANT in finding relevant anchors automatically.

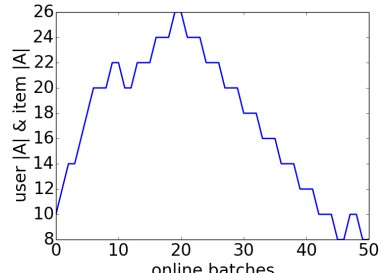

Figure 8: Using NBANT for online learning increases $|A|$ with more data before natural clusters emerge, and $|A|$ decreases to a similar value (8) as the non-online version in Table 3.

## L    EFFECT OF $\lambda_1$ AND $\lambda_2$

In this section we further study the effect of the hyperparameters $\lambda_1$ and $\lambda_2$. Recall that our full objective function derived via small variance asymptotics is given by:

$$\min_{\substack{\mathbf{T} \geq 0 \\ \mathbf{A}, \theta, K}} \sum_i D_\phi(y_i, f_\theta(x_i, \mathbf{TA})) + \lambda_2 \|\mathbf{T}\|_0 + (\lambda_1 - \lambda_2)K \tag{36}$$

The role of hyper-parameter $\lambda_2$ is clear: For a fixed $\lambda_1$ and integer valued variable $K$, tuning $\lambda_2$ controls the trade-off between sparsity of $\mathbf{T}$ and model performance (see Table 13).

The role of hyper-parameter $\lambda_1$ is more subtle. It can be considered as a weighing fraction for scalarizing an underlying multi-objective optimization problem. To elaborate, one can consider our goal as a multi-objective problem of minimizing the predictive loss while simultaneously using a minimal number of anchors ($K$). Then the hyperparameter $\lambda_1$ can be used to select a solution along the Pareto front. In other words, tuning the hyperparameter $\lambda_1$ allows us to perform model selection by controlling the trade-off between the number of anchors used and prediction performance. We apply eq (36) on the trained models in Table 2 and report these results in Table 15. Choosing a small $\lambda_1 = 2 \times 10^{-5}$ selects a model with more anchors ($|A| = 1,000$) and better performance (ppl = 79.4), while a larger $\lambda_1 = 1 \times 10^{-1}$ selects the model with fewest anchors ($|A| = 100$) with a compromise in performance (ppl = 106.6).

Table 15: An example of model selection on the trained language models using LSTM trained on PTB. Tuning the hyperparameter $\lambda_1$ and evaluating eq (36) allows us to perform model selection by controlling the trade-off between sparsity as determined by the number of anchors used and prediction performance.

| Method | $|A|$ | Init $A$ | Ppl | # nnz($\mathbf{T}$) | $\lambda_1$ | $\lambda_2$ | eq (36) |
|---|---|---|---|---|---|---|---|
| | 2,000 | Dynamic | 77.7 | 245K | $2 \times 10^{-5}$ | $1 \times 10^{-6}$ | 4.64 |
| ANCHOR & TRANSFORM 200 | 1,000 | Dynamic | 79.4 | 214K | $2 \times 10^{-5}$ | $1 \times 10^{-6}$ | **4.61** |
| | 500 | Dynamic | 84.5 | 171K | $2 \times 10^{-5}$ | $1 \times 10^{-6}$ | 4.62 |
| | 100 | Dynamic | 106.6 | 25K | $2 \times 10^{-5}$ | $1 \times 10^{-5}$ | 4.92 |
| | 2,000 | Dynamic | 77.7 | 245K | $1 \times 10^{-4}$ | $1 \times 10^{-6}$ | 4.80 |
| ANCHOR & TRANSFORM 200 | 1,000 | Dynamic | 79.4 | 214K | $1 \times 10^{-4}$ | $1 \times 10^{-6}$ | 4.69 |
| | 500 | Dynamic | 84.5 | 171K | $1 \times 10^{-4}$ | $1 \times 10^{-6}$ | **4.66** |
| | 100 | Dynamic | 106.6 | 25K | $1 \times 10^{-4}$ | $1 \times 10^{-5}$ | 4.93 |
| | 2,000 | Dynamic | 77.7 | 245K | $1 \times 10^{-1}$ | $1 \times 10^{-6}$ | 204.6 |
| ANCHOR & TRANSFORM 200 | 1,000 | Dynamic | 79.4 | 214K | $1 \times 10^{-1}$ | $1 \times 10^{-6}$ | 104.6 |
| | 500 | Dynamic | 84.5 | 171K | $1 \times 10^{-1}$ | $1 \times 10^{-6}$ | 54.6 |
| | 100 | Dynamic | 106.6 | 25K | $1 \times 10^{-1}$ | $1 \times 10^{-5}$ | **14.9** |

