# OpenReview forum: "Anchor & Transform: Learning Sparse Embeddings for Large Vocabularies"
_ICLR.cc/2021/Conference — ICLR 2021 Poster_

### Official Review · AnonReviewer4 · 2020-10-28
**An ingenious two-step method on representation learning**

**Rating:** 7
**Confidence:** 4

**Review:**

In this paper, the authors proposed a method to learn efficient representations of discrete tokens. They took a two step approach: in step 1, they learn "full fledged" embeddings for a subset of anchor tokens. In step 2, they learn a sparse matrix that is used to relate all tokens to the set of chosen anchors. This two-step approach reduced the overall number of parameters. The sparse matrix T can also encode domain knowledge (e.g. knowledge graphs). In the experiment section, the authors showed that their approach has good performance on several language tasks, with far fewer parameters.

In general the paper is well written and the flow is easy to follow. I find the main idea plausible and ingenious. For language tasks and word embeddings, anchoring method has been shown to be effective in several tasks already (e.g. [1] http://papers.nips.cc/paper/8152-the-global-anchor-method-for-quantifying-linguistic-shifts-and-domain-adaptation). The authors took two steps forward: 1) instead of in [1] where the entire vocab is used for anchoring purposes, the authors used a subset of tokens which reduces the amount of parameters. 2) they use a sparse T matrix to relate other tokens to anchors which again has reasonable prior: the meaning of a word can be efficiently defined by a few good chosen anchors. Although this paper is probably related to other strains of research (e.g. leaning manifolds for IR/NLP where anchoring is also a key concept, which the authors could have admittedly surveyed more), I particularly liked the fact that the two-step procedure decomposes two tasks that are often mixed together for embedding tasks: learning representation vs learning relations.

While the authors claimed that they can further impose domain knowledge in the learning process (which I think this is at least a good attempt), this part in general feels a bit less convincing. To be specific, there can be a variety of knowledge (like related, is a subset of, analogy, etc.). It is not clear how the distinction of different types of knowledge can be incorporated. What the authors proposed is lumping them into the notion of "positive pair" and relax constraints on them. This may or may not suffice (for the purpose of adding domain knowledge), but on paper, there is a chance that some finer structures of the domain knowledge may get lost. It's not clear how much gain (especially the experiment section, for fair comparison purposes where other methods do know use domain knowledge in particular) is from incorporating domain knowledge; an ablation study might help.

Another question is about training. It's not obvious to me how to guarantee that for every row of T, there is at least 1 non-zero element. Is some specific tuning needed for rows corresponding to rare words? How the regularization strength $\lambda$ on T is selected?

The reduction of parameters is while keeping task performance is illustrated quite well in the experiment section. Their method does not reduce the theoretical complexity (still linear w.r.t. vocab size, as T must have at least one element per row), but in practice the reduction (which mostly comes from savings of dimensionality) is quite obvious.

---

> ### Author Response · Authors · 2020-11-20
> **Author response**
>
> Thank you for your constructive comments, feedback, and for finding the method to be ingenious. We answer your questions here:
>
> [R4 domain knowledge] We do not use any domain knowledge in the experiments reported in the main paper. Experiments with domain knowledge are presented in Appendix, e.g. Table 10, 11 and we show further improvements using WordNet and co-occurrence statistics to impose relationships between the anchor and non-anchor words, particularly on compression. We agree with the reviewer that there can be more elaborate manners to incorporate domain knowledge - we simply attempted to give several proposals as a proof-of-concept that achieves promising results. These more elaborate methods are ripe areas for future research.
>
> [R4 non-zero rows] That is a good question. We didn’t do any specific tuning for rare words/items and consider it as a feature, not a bug. It shows the method adaptively eliminates only undesired words. Empirically we found all words in WikiText-103 to have non-zero rows when doing little compression (38.4 perplexity with 6.5M parameters). For more compression (54.2 perplexity with 0.4 parameters), 179K words (out of 267K total) had zero rows. **ANT clearly differentiates between useful words to model and rare words that it can ignore for compression:**
>
> Zero rows: Anarky, Perl, Voorhis, Gaudí, Lat, Bottomley, Nescopeck, Tzara, LeMond, Doby, Gulfton, Mileena, Kuznetsova, FN, Wallenberg, Ahalya, Betances, Cabral, Braham, Coatbridge, rajah, Finkelstein, Wilders, Swaminarayan, Youkilis, Robben, Satin, Whorf, Ralphie, BN, Betelgeuse, Vithoba, FIU, Astatine, Hextall, Massino
>
> Non-zero rows: out, about, than, game, between, later, three, most, while, new, On, made, film, such, season, where, before, years, only, 2, up, He, they, after, would, time, into, It
>
> We also noticed that certain rare words that might be predictive are assigned non-zero rows in T, such as:
> sociologists, losers, finder, deadlines, indestructible, causeways, captions, outsourced, rotors, refrigerated, reconsiders, glacially, heartening, unchallenging, roughest
>
> We believe the objective function learns to make all entries 0 in a row only if including a few doesn’t increase the log-likelihood much, i.e. an all-zero vector can behave like the embedding of a UNK token and having a UNK token is not much worse for such rare words.
>
> We observed a similar trend in MovieLens rating prediction. Only 2673 out of 59047 movies had an entire zero row, **of which 84% only had 1 rating (i.e. very rare movies).** Furthermore, all user transformations had non-zero rows - on average each user transformation had 3 (out of 8) non-zero transformations wrt user anchor embeddings. **We added a discussion on this in section 4.5 under ‘4) Zero transformations learned’.**
>
> [R4 lambda] The regularization parameter \lambda is selected via grid search on a held-out validation set.

---

### Official Review · AnonReviewer3 · 2020-10-29
**A neat idea to pose sparse embedding model as a sparse linear combination of dense latent vectors. Nice interpretation as a Bayesian prior. Good empirical evidence on multiple NLP and Information Retrieval tasks. Might need comparison against some other sparse models like Compostional Embeddings, SNRM, SOLAR.**

**Rating:** 6
**Confidence:** 4

**Review:**

This paper proposes ANT to solve the problem of learning Sparse embeddings instead of dense counterparts for tasks like Text Classification, Language Modeling and Recommendation Systems. When the vocabulary size |V| runs into several 100Ks or millions, it is impractical to store one dense vector per label. Hence the paper proposes to only store a few anchor/latent vectors (the matrix is A with |A|<<|V|). All label vectors are expressed as linear combinations of a 'few' anchor vectors. To train this end-to-end, we need a transformation matrix T such that T*A = E (E is V\times d embedding matrix). T has to be structured, i.e., each row of T has to be sparse and positive only (although negative weights are also fine, I'm not sure if weight redundancy is that important).

This pipeline is trained end to end using YOGI optimizer for regular gradient updates and 'proximal gradient descent' for T which does soft thresholding with a lower bound of 0 (accomplishing both sparsity and positivity part).

This design admits multiple ways of initializing the anchors A. And the authors perform experiments with both frequent token vectors and random anchor vectors (both have their merits, random seems to be a robust choice).

The authors provide a statistical interpretation of their approach using a generative formulation to the embedding vectors in terms of the latent vectors (using a Indian Buffet Process membership matrix Z).

The experiments span two major domains, NLP and Information Retrieval. Across multiple NLP datasets, ANT outperforms Sparse-Coding (Chen et. al. 2016)  and Post-Sparse-Hash (Guo et.al. 2017). On the IR task with MovieLens dataset, the primary comparison is against SLIMMING (Liu et.al. 2017). While gains are substantial on the NLP tasks, they seem minimal on the MovieLens task.

I've listed most pros above. The cons are here:
1. The idea seems a little similar to Compositional Embeddings (Shi et. al. 2020, Ginart et. al.2019). It might warrant a discussion or comparison.
2. There are other sparse embedding methods like SNRM (Zamani et.al. 2018) and SOLAR-Sparse Orthogonal ...(Medini et.al. 2020) which might be comparison candidates at-least for IR tasks.
3.  The precision in table 1 for ANT anomalously increases when |A| is reduced. ANy explanation as to why this happens? The information bottleneck is supposed to reduce precision right?

---

> ### Author Response · Authors · 2020-11-20
> **Author Response**
>
> Thank you for your constructive comments and feedback. We answer your questions here:
>
> [R3 baselines 1] MixedDim Embeddings by Ginart et al., (2019) are specifically designed for recommender systems and we did compare to MixedDim in our MovieLens experiment in Table 3, where we showed improvements. MixedDim also requires a significant preprocessing time to count the frequency of users and items that scales poorly to very large numbers of users and items. Compositional Embeddings by Shi et al., (2020) is a very similar paper from the same authors as Ginart et al., (2019) which we compared to.
>
> [R3 baselines 2] SNRM seems to be orthogonal to our focus. It is designed to yield sparse embeddings for documents, whereas we operate at sparsifying the word embedding table. At token level SNRM still requires the full embedding table (as described on page 4 of their paper) and thus has no parameter size reduction with respect to vocabulary size. In fact, it is possible to use ANT in combination with SNRM which we believe can inspire future work in combining local (word/object-level) and global (document-level) compression.
>
> Thanks for pointing out about SOLAR. We were not aware of the paper and it seems to not be published anywhere and was only released on arxiv 2 weeks before the ICLR deadline.
>
> [R3 AG news] AG News is the smallest dataset in our experiments and large models with full embedding matrices are likely to overfit and generalize poorly. ANT with low-rank anchor embeddings and sparse transformations seems to provide a regularization effect that learns better word representations and generalizes better. We did not observe this for the larger datasets for text classification (DBPedia, Sogou-News, and Yelp-review, see Table 10 in the Appendix), language modeling (PTB and WikiText-103, see Table 2), and recommender systems (MovieLens in Table 3), where there was a clear trade-off between performance and compression.

---

### Official Review · AnonReviewer2 · 2020-11-01
**This paper proposes a practical solution to cut the embedding storage. But the Bayesian interpretation is not persuasive and there are still missing pieces in the experiments.**

**Rating:** 5
**Confidence:** 4

**Review:**

This paper introduces a row-rank approximation of embeddings using “anchors”. It also proposes a probabilistic interpretation of their method as a non-parametric Bayesian dictionary learning model, which can be inferred by optimizing the small-variance asymptotic objective.

What I agree with the authors are:
i) Using properly chosen basis vectors may greatly reduce the memory cost for embeddings, especially for huge vocabulary sizes (e.g. over 100 million).
ii) The initialization for basis vectors is extremely important and should be updated through training.
iii) Experimental results look reasonable in this paper.

What I feel confused about are:
i) Why interpret this method in a Bayesian non-parametric way? To be more specific:
i.1) The final objective function (5) does not involve Bayesian posterior inference. If you want a point estimation of sparse representation + learnable anchors, you don’t need a Bayesian model.
i.2) Bayesian non-parametric is useful because it can automatically learn the model size. In your case, |A|. You mention this point in Figure 3, but there is no online learning result showing that your model has the capacity to grow the model through training. One example is “Truly Nonparametric Online Variational Inference for Hierarchical Dirichlet Processes” by M. Bryant and E. Sudderth.

ii) The vocabulary size in your experiment is decent, but not very big. Normally in a recommendation system, the vocabulary size can be the number of users, which is at least 100M. Normally the embedding size is around 16 to 64 in real systems. The proposed method could be a huge gain in storing such a huge embedding table. But I cannot see an experiment at this vocabulary level. Even rough results at this level could make this paper much stronger.

iii) This is a minor point, but AUC results in MovieLens besides MSE can reflect the ranking quality in recommendations.

Overall, this paper proposes a practical solution to cut embedding storage. But the Bayesian interpretation is not persuasive and there are still missing pieces in the experiments.

---

> ### Author Response · Authors · 2020-11-20
> **Author Response**
>
> Thank you for your constructive comments and feedback. We answer your questions here:
>
> [R2 Bayesian] We provide a statistical interpretation as a Bayesian nonparametric prior since it shows us a way to automatically learn the optimal number of anchors. Empirically, in Figure 4, we plot the value of eq (5) across values of |A| after a comprehensive hyperparameter sweep on ANT across 1000 settings. In comparison, nbANT optimizes |A| and reaches a good value of eq (5) from just **one**run, which shows that **the Bayesian interpretation provides successful principles for algorithm design to further reduce training efficiency.**
>
> Furthermore, it is not true that Bayesian inference must always be associated with a posterior distribution - in fact, several papers before us have applied Small Variance Asymptotics to obtain approximate point estimates for hierarchical Bayesian models (Broderick et al., 2013a; Jiang et al., 2012; Roychowdhury et al., 2013). In our case, the quantities of interest in Bayesian inference are the anchor matrix A and transformation matrix Z, which are very high-dimensional matrices. Modeling the posterior distribution is useful for scalar quantities, e.g. to obtain confidence intervals, but not so much for high-dimensional latent variables (Zaheer et al., 2016, Tristan et al., 2015). Furthermore, we do only care about a point estimate for |A| (and A and Z) so we do not actually need a posterior distribution.
>
> Zaheer et al., Exponential Stochastic Cellular Automata for Massively Parallel Inference. AISTATS 2016
>
> Tristan et al., Efficient Training of LDA on a GPU by Mean-for-Mode Estimation. ICML 2015
>
> [R2 online] Our Algorithm 2 in the appendix describes the procedure for the model to grow/contract in terms of the number of anchors, |A|, during training. We agree that online settings are important and **we convert Algorithm 2 to an online version**inspired by the paper you mentioned. Instead of looping through all mini-batches for all epochs, we treat each batch as a new set of data coming in an online setting and train on that batch until convergence, modify |A| as in Algorithm 2, before moving onto the next batch. In this significantly more challenging online setting, nbANT is able to learn well and **achieve an MSE of 0.875 with 1.25M non zero parameters.** Initially, the clusters grew steadily from 10 up to 26 as online batches were seen. As even more batches were seen, the number of anchors decreased steadily from 26 to 10 and oscillated between 8 and 10. This means initially some connections/groupings between online batches were not clear, but with more data, natural clusters merged together. Interestingly this online version of nbANT settled on a similar range of final user (8) and item (8) anchors as compared to the non-online version (see Table 3), which confirms the robustness of nbANT in finding relevant anchors automatically for growing model capacity. **We added these results under ‘nbANT’ in section 4.3 movie recommendation and in Appendix K.3.**
>
> [R2 large vocab size] Thank you for suggesting this experiment. To the best of our knowledge, we could not find public datasets with more than 100M users. If you have one in mind, please let us know. In the meantime, we are currently experimenting with a much larger Amazon Review dataset (https://nijianmo.github.io/amazon/index.html; Ni et al., 2019) which is the largest public recommendation system benchmark so far containing 233M reviews spanning 43.5M items and 15.2M users. Even for a dataset of this size, there are no recommender system baselines on the entire dataset, they all take a subset of product categories (e.g. Amazon clothing). We will update this comment and the paper with new results by the weekend.
>
> Ni et al., Justifying recommendations using distantly-labeled reviews and fined-grained aspects. EMNLP 2019
>
> [R2 AUC] We agree that AUC scores are very useful for evaluating ranking tasks, but these are only applicable for MovieLens if explicitly converted to a ranking task. We performed the rating prediction task as measured using MSE which has been the precedent for this dataset: recent works using MovieLens for recommender systems have all studied rating prediction as measured using MSE (Ginart et al., 2019, Strub et al., 2017).
>
> Strub et al., Hybrid Recommender System based on Autoencoders. 2017
>
> Saadati et al., 2019. Movie Recommender Systems: Implementation and Performance Evaluation. 2019
>
> Ginart et al., Mixed Dimension Embeddings with Application to Memory-Efficient Recommendation Systems. 2020

---

> ### Author Response · Authors · 2020-11-24
> **Scaling up to Amazon Product reviews - 43.5M items and 15.2M users**
>
> [R2 large vocab size] We have performed additional experiments with a much larger Amazon Review dataset (https://nijianmo.github.io/amazon/index.html) which is the largest public recommendation systems benchmark so far containing 233M reviews spanning 43.5M items and 15.2M users. We first experiment on a commonly used subset of the data, Amazon Electronics, to ensure our results match published baselines (Wan et al., 2020).
>
> 			User |A|	Item |A| 	MSE		  nnz params (M)
> 	MF	  9.84M	   0.76M		1.524		170
> 	ANT	 20		  8			**1.422**	25.8
> 	ANT	 8		   3			1.529		7.10
> 	ANT	 5		   3			1.591		**3.89**
>
> Next, we scale our experiment to the entire dataset:
>
> 			User |A|	Item |A| 	MSE		  nnz params (M)
> 	MF	  43.5M	   15.2M		1.164		939
> 	ANT	 15		  10		   **1.099**	201
> 	ANT	 8		   8			1.167		**95.9**
>
> For both, we find that ANT compresses embeddings by **25x** on Amazon Electronics while maintaining performance, and **10x** on the full Amazon reviews dataset. **We added these results in section 4.4 product recommendation.**
>
> Wan et al., Addressing Marketing Bias in Product Recommendations. WSDM 2020

---

### Decision · Program_Chairs · 2021-01-07
**Final Decision**

**Decision:**

Accept (Poster)

**Comment:**

This paper proposes a method to cope with large vocabulary sizes. The idea is to find a small number of anchor words and to express every other word as a sparse nonnegative linear combination of them. They give an end-to-end method for training, and give a statistical interpretation of their algorithm as a Bayesian nonparametric prior (in particular an Indian restaurant process). They give extensions that allow them to deduce the optimal number of anchors which allows them to avoid needing to tune this hyperparameter. Finally they give a variety of experiments, particularly in language and recommendation tasks. The results on language are particularly impressive, and in the author response period, at the behest of a reviewer, they were able to extend the experiments to the Amazon Review dataset which contains 233M reviews on 43.5 M items by 15.2 M users.

This paper is a nice combination of a simple but powerful idea, and a range of experiments demonstrating its utility. Other papers have proposed related ideas, but here the main novelty is in (1) using a small number of anchors that can incorporate domain knowledge and (2) using a sparse linear transformation to express other words in this basis. One reviewer did not find the Bayesian nonparametric interpretation to be fruitful, since it does not lead to techniques for handling growing datasets (e.g. if the ideal number of anchors changes over time).